# The distribution pattern and temporal variation of desert riparian forests and its influencing factors in the downstream Heihe River Basin, China

Jingyi Ding[1], Wenwu Zhao[1,2]*, Stefani Daryanto[2], Lixin Wang[2]*, Hao Fan[1], Qiang Feng[1,3] , Yaping Wang[1]

[1]State Key Laboratory of Earth Surface Processes and Resource Ecology, Faculty of Geographical Science, Beijing Normal University, Beijing 100875, P. R. China

[2] Department of Earth Sciences, Indiana University-Purdue University, Indianapolis (IUPUI), Indianapolis, Indiana 46202, USA

[3]College of Forestry, Shanxi Agricultural University, Taigu, Shanxi 030801, P. R. China

*Correspondence to*: W. W. Zhao (Zhaoww@bnu.edu.cn) and L. X. Wang (lxwang@iupui.edu)

**Abstract.** Desert riparian forests are the main restored vegetation community in the Heihe River

Basin. They provide critical habitats and a variety of ecosystem services in this arid environment.

Since they are also sensitive to disturbance, examining the distribution pattern, temporal variation of

desert riparian forest and their influencing factors are important to determine the limiting factors of

vegetation recovery after long-term restoration. In this study, field experiment and remote sensing

data were used to determine the spatial and temporal pattern of desert riparian forests and their

relationship with environmental factors. Across different distance from the river channel, we

classified five types of vegetation communities. Community coverage and diversity formed bimodal

pattern peaked at the distance of 1000 m and 3000 m from the river channel. In general, temporal

NDVI trend was positive across different distances from the river channel, except for the region

closest to the river bank (i.e. within 500 m from the river channel), which already underwent

degradation since 2011. Spatial heterogeneity of soil properties (e.g. soil moisture, soil physical

properties and soil nutrition) and temporal variation of water availability (e.g. annual average and

annual variability of groundwater, soil moisture and runoff) explained 74% of the vegetation variance.

Spatial heterogeneity factors, accounting for 98.4% of the total variance explained, positively

influenced the community diversity, structure, average NDVI and change variability of NDVI trend.

Temporal variation factors accounting for 35.9% the total variance explained, positively influenced

the community density and average NDVI. With surface (0-30 cm) and deep (100-200 cm) soil

moisture, bulk density and annual average of 100 cm soil moisture regarded as major determining

factors of community distribution and temporal variation, conservation measures that protect soil

structure and prevent soil moisture deficiency (e.g., artificial soil cover and water conveyance

channel) are suggested to better protect desert riparian forests under climate change and intensive

human disturbance.

## 1 Introduction

Riparian zone is the linkage between terrestrial and aquatic ecosystem (Naiman and Décamps, 1997),

which plays an important role in ecological processes and provides a variety ecosystem services, such

as sand stabilization and carbon sequestration (Naiman et al., 1993; Décamps et al., 2004). Desert

riparian forests, also known as 'Tugai forests', are considered as the main body of riparian zone in the
hyperarid areas, mainly located in the floodplains of the major Central Asian rivers (Gärtner et al.,
2014). They provide critical habitats for various species and function as the "ecological shelter" against
desertification in the hyperarid area (Thevs, 2008; Ding et al., 2016). However, due to the low diversity
level and weak resilience, desert riparian forests are sensitive to disturbance and likely to be threatened
by desertification under changing environment (Ling et al., 2015; Li et al., 2013).
Desert riparian forests are the main communities in the Heihe River Basin, the second largest
inland river in China (Feng et al., 2015). During the past century, human population increase and
overexploitation of the upstream water resources led to significant degradation of the downstream
desert riparian forests (Wang et al., 2014). Since 2000, ecological water conveyance project (EWCP), a
restoration project aimed to deliver water downstream has been implemented to restore the ecosystems
of the Heihe River Basin (Yu et al., 2013). Every year, about 300 billion $m^3$ of water were delivered
using concrete channels, built perpendicular to the river aiming to expand the river impact and to
deliver water for irrigation. While most downstream vegetation has been restored (Wang et al., 2014;
Lü et al., 2015), nearly 20% of the oasis area covered by desert riparian forests still underwent major
degradation in spite of the rising groundwater level and better downstream water condition (Zhang et
al., 2011a; Lu et al., 2015). To conserve and restore this fragile ecosystem more effectively, studies that
address the variation of desert riparian forests and their relationship with the environmental factors
need to be conducted.
The distribution pattern of desert riparian forests is the result of long-term interaction between
vegetation and multiple environmental factors, particularly water availability (Goebel et al., 2012; Li et
al., 2013). With river acting as the main supply of water in desert riparian forests, the distance from
river channel could be regarded as a proxy to water availability (including groundwater), which
declined with the weakening of river influence (Hao et al., 2010; Chen et al., 2014). Species diversity
would peak where groundwater depth was around 2-4 m, before it started to decrease once groundwater
went below 4-4.5m and deficiency in soil moisture occurred (Zheng et al., 2005; Li et al., 2013). While
this could be the case for some hyperarid zones (e.g., Tarim river) where groundwater dropped rapidly
away from the river bank to about 6 m deep at the distance of 1000 m from river channel (Aishan et al.,
2013), the groundwater table remained above 4 m at the distance of 3800 m from the Heihe river
channel (Wang et al., 2011; Fu et al., 2014). Yet some sites were not completely restored at the Heihe

riparian zones and the downstream vegetation community still shifted from multiple layers of trees to shrubs (He and Zhao, 2006; Zhang et al., 2011a). Previous study by Zhu et al. (2013) showed that Patrick's richness index and Shannon–Wiener's index of downstream vegetation formed a bimodal pattern along groundwater depth in the Heihe River Basin, indicating that there could be other factors affecting the distribution of desert riparian forests.

Apart from water, soil properties, such as soil moisture, soil physical and soil chemical properties also shape the community characteristics by influencing the ecological and hydrological process (Stirzaker et al. 1996; Salter and Williams, 1965). Soil moisture, influenced by precipitation and groundwater, is the direct water source for the desert riparian forests (Wang et al., 2012). Interactions between communities and extreme environmental stress could cause non-unimodal responses in the hyperarid zone (Oksanen and Minchin, 2002), although other study in semiarid zone showed a unimodal pattern (Li, 2006; Hao et al., 2010; Li et al., 2013). With different depth of soil moisture exerted different impacts on vegetation (D'Odorico et al., 2007; Fang et al., 2016), the decline of soil moisture would reduce the abundance of tree and herb species, resulting in the community shift to drought-tolerant vegetation types along the distance from river channel (Zhu et al., 2014). Some studies also found that the heterogeneity in soil properties was the reason for the evolution of dominant species in arid area and the changes in soil nutrients contribute greatly to species diversity (DíAz and Cabido 2001; Yang et al. 2008).

As desert riparian forest is the main community that maintains the ecosystem function in hyperarid zone, comprehensive research on the spatial and temporal variation of the vegetation will benefit restoration of the whole area. Spatial distribution and temporal variation of vegetation can reflect how communities respond to the changing environment during ecological restoration (Bakker et al., 1996; Scott et al., 1996). Although variation of vegetation characteristic during restoration process and its relationship with runoff and groundwater have been addressed in previous studies by using large scale dataset (e.g., MODIS-NDVI, SPOT-NDVI) (Jia et al., 2011; Wang et al., 2014; Geng et al., 2014), they only captured the general trend of the whole study area rather than focusing on the desert riparian forest. More importantly, their data resolution could not accurately delineate the temporal variation pattern at different distances from river channel. Currently, there have been limited number of studies that tried to disentangle the impacts of spatial heterogeneity and temporal variation factors on the vegetation communities (Zhu et al., 2013; Xi et al., 2016) due to the lack of long term monitoring data, inhibiting

the effective restoration of desert riparian zone.

2        In this research, we aim to explore the impacts of those aforementioned factors and to examine the

distribution pattern and temporal variation of vegetation communities in the Heihe desert riparian forest.
We investigated variability in desert riparian forests sites that are differently located along the
perpendicular direction from the river channel. Changes of floristic composition, community structure
and diversity were used to depict community distribution pattern, and the variation of NDVI at each
gradient from 2000-2014 was used to depict the temporal variation. Spatial heterogeneity factors (e.g.,
soil moisture, soil physical properties and soil nutrition) and temporal variation properties (e.g., annual
average and annual variability of groundwater, soil moisture and runoff) were used to explain the
vegetation community variance. The objectives of this study were to: (1) explore the distribution
pattern of desert riparian forest along the perpendicular direction from the river channel and the
temporal variation of NDVI in desert riparian forest since 2000, (2) analyze the effect of spatial
heterogeneity factors and temporal variation factors on the community characteristics of desert riparian
forests, and (3) explore the community resilience of desert riparian forest along the distance from river
and suggest suitable restoration and protection measures for desert riparian forests under changing
environment.
**2 Data and methods**
**2.1 Study area**
The study was conducted in the downstream Heihe River (40°20′–42°30′N; 99°30′–102°00′E), in the
Ejina Oasis, Inner Mongolia, Northwest China. The oasis covers an area of $3 \times 10^4$ km$^2$, with declining
surface elevation (i.e., 1127 m to 820 m above sea level) from the southwest to the northeast (Qin et al.,
2012). This region has a typical continental arid climate with mean annual temperature of 8.77 ℃. Its
maximum and minimum temperatures usually occur in July (41°C) and January (−36 ℃) (Wen et al.,
2005). The mean annual precipitation is <39 mm, 84% of which occurs during the growing season
(May to September), while the mean annual potential evaporation is >3,390 mm (Chen et al., 2014).
Prevailing wind direction is northwest, mean annual wind velocity is 2.9-5.0 m s$^{-1}$, and annual number
of gale (>8 m s$^{-1}$) days is 70 days or so (Chen et al., 2014).
The Heihe River originates from rainfall and snow melt in the Qilian Mountains. It branches into
the Donghe River and the Xihe River at Langxinshan Mountain and ultimately flow into the East Juyan
Lake and the West Juyan Lake in Ejina. The population in the Ejina oasis is 32,410 (Ejina statistical
office, 2012). The local economy mainly depends on the cantaloupe plantation and animal husbandry
(e.g. sheep, cattle and camel). Ejina Oasis is one of China's most important tourist attractions with
respect to desert riparian forests, attracting almost 200,000 visitors per year during September to
October (Hochmuth., 2014). Two primary roads are built parallel to the river channel and across the
south of the oasis respectively, mainly used for transportation and traveling.
Due to sparse precipitation and hyperarid environment, Heihe River is the main source of recharge
for the groundwater system in Ejina Oasis (He and Zhao, 2006). As the distance from river channel
increases, water availability declines and the vegetation shifts from desert riparian forests to desert
scrub. The desert riparian forests are the main components of Ejina oasis. They mainly grow along the
river banks and spread across the fluvial plain, with the dominant vegetation including *Populus*
*euphratica, Tamarix ramosissima, Lyceum ruthenicum, Sophara alopecuriodes, Karilinia caspica,* and
*Peganum harmala* (Zhao et al., 2016). The sparse and drought tolerant desert species such as
*Reaumuria soongorica*, *Zygophyllum xanthoxylon* and *Calligonum mongolicunl* are mainly distributed
in the Gobi desert. The main soil types in the area are shruby meadow soil, aeolian soil and grey-brown
desert soils. Saline-alkaline soils and swamp soils also exist in the lake basins and lowlands (Chen et al.,

19   2014).

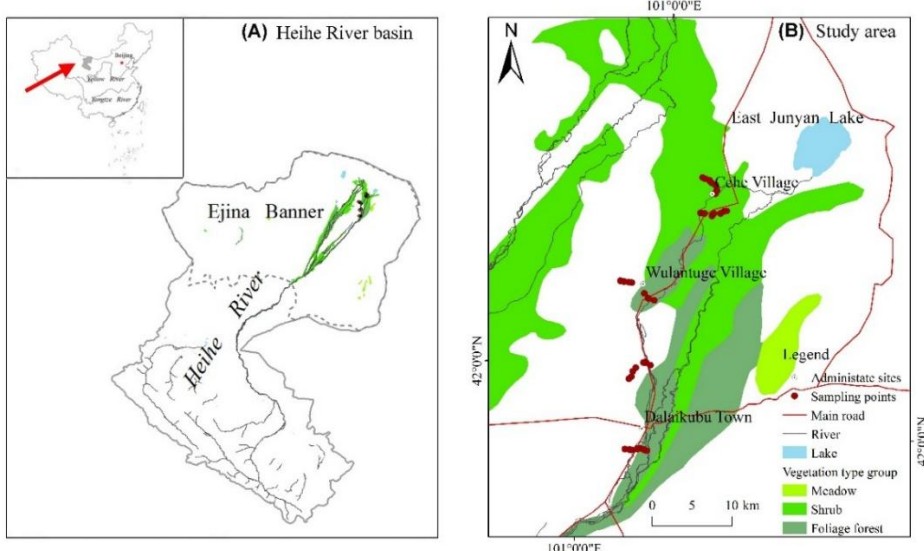

**Figure 1.** The Heihe River basin in China (A) and the location of sampling points in the study area
(B). Two primary roads are built parallel to the river channel and across the south of the oasis,

1       respectively.

**2.2 Spatial field survey and experimental design**
In the downstream Heihe River Basin, the desert riparian forest makes up the main body of the desert
oasis, mainly comprised of tree, shrub and grass communities. The forests are distributed along the
Heihe River from 0 m to 2000 m from river channel (Si et al., 2005; Guo et al., 2009). However, the
spatial extent of the riparian zone is difficult to be delineated due to the heterogeneity of landform
mosaics (Décamps et al., 2004). Therefore our study covered a length across 3000 m distance from the
Heihe river channel to fully cover the distribution pattern of its desert riparian forests.
Our field survey was conducted in July 2015, after the ecological water conveyance delivered. The
ecological water conveyance is implemented according to the water dispatching scheme and conducted
in the April, July, August, September and November with scheduled discharge (Feng et al., 2015). Five
transects perpendicular to the river were selected randomly as replicates under the premise of
consistency in soil type and micro-topography. Due to the regulated water discharge, the ecological
water conveyance only affects the sites near the river bank (within 100 m radius) (Liu et al., 2008).
Therefore, vegetation and soil samplings were conducted perpendicular to the river channel and the
distance from the river channel was stratified into seven gradients: 100 m, 500 m, 1000 m, 1500 m,
2000 m, 2500 m, and 3000 m, respectively, generating a total of 35 sampling sites. Those sites were far
from farmlands, irrigated channels and reservoirs to minimize the impact of human disturbance and
other water resources. Although, there is a main road extending across the oasis and almost parallel to
the river channel (Fig. 1), the vegetation community growing nearby the road is considered to be
undisturbed by the road as the road is separated from the surroundings by iron wire.
Three tree quadrats (30 m $\times$ 30 m) and shrub quadrats (10 m $\times$ 10 m) were established in each site.
The number of each species (tree and shrub), plant height, coverage and diameter at breast height
(DBH) of the trees ($\geq$ 2 m) were recorded individually. Four (2 m $\times$ 2 m) herb quadrats were
established at each corner of the tree or shrub quadrat to collect data on the number of plants,
vegetation cover and height.
At each site, soil samples and soil moisture samples were randomly collected in three replicates
using auger (5 cm in diameter). Soil gravimetric water content (SWC) was collected at depths of 5, 10,
15, 20, 30, 40, 50, 60, 70, 80, 100, 120, 140, 160, 180, 200 cm, weighed at the time of sampling as well
as after oven drying at 105 ℃ for 48 hours. At some sites where groundwater was less than 2 m, the
SWC sampling stopped at the depth of groundwater table. Bulk density (BD) was measured by
collecting undisturbed soil cores at surface layer using a stainless-steel cutting ring (100 cm$^3$ in volume)
with three replicates each site and oven dried at 105 ℃ until they reached constant weight. Soil particle
size distribution and soil chemical properties (soil organic matter, total nitrogen, total phosphorus and
total salt content) were analyzed in the laboratory using 0-100cm soil samples that were collected
separately in each site.
**2.3 Temporal data collection and processing**
In order to analyze the long term vegetation variation since the implementation of ecological water
conveyance, we analyzed NDVI data from 2000 to 2014. As the NDVI measures vegetation status,
including coverage and vigor, we used the maximum NDVI during growing season as the indicator of
vegetation community characteristics. The maximum NDVI during growing season (May-October)
generally indicated the best vegetation state of the whole year (Wang et al., 2014). The NDVI in each
sampling site during 2000-2014 were calculated using ENVI (5.0) based on the Landsat TM/ETM
image (30 m) acquired from Geospatial Data Cloud (http://www.gscloud.cn/). The variable
environment factors such as 2 cm soil moisture, 100 cm soil moisture and groundwater in each site
during the research period were extracted from the retrieved remote sensing data with 1000 m
resolution (Zeng et al., 2016). Land use change information from 2000-2014 was extracted from land
use data at a scale of 1:100,000 (for 2000 and 2014) (Liu et al., 2002; Zhong et al., 2015). The diurnal
and annual variation of soil moisture were depicted by the monitoring data of soil moisture from
2013-2015 (Liu et al., 2011; Li et al., 2006). The retrieved remote sensing data, monitoring data and
land use data were acquired from Environmental & Ecological Science Data Center for West China,
National Natural Science Foundation of China (http://westdc.westgis.ac.cn). Runoff data at Zhengyixia,
a hydrological station at the border of the downstream Heihe, was collected from the Hydrological
Almanac of China from the Chinese Academy of Sciences.
**2.4 Statistical analysis**
The P (importance value) of each tree, shrub and herb in each plant site was calculated for each species
using the following formulas (Zhang and Dong, 2010):
$$\text{P}_{\text{Tree}} = (RDen + RDom + RH)/3 \qquad (1)$$
$$\text{P}_{\text{Shrub or Grass}} = (RDen + RDom + RC)/3 \qquad (2)$$
where $RDen$ is the relative density, $RDom$ is the relative dominance, $RH$ is the relative coverage
and $RC$ is the relative coverage.
In our study, the total diversity index of community was deployed to depict the community
diversity in each site. According to the characteristic of community vertical structure, the total diversity
index of community is measured using the weight of indices in different growth types. The weight is
the average of the relative coverage and the thickness of the leaf layer (Fan et al., 2006). We applied
the following formula (Gao et al., 1997):
$$W_i = (C_i/C + h_i/h)/2 \qquad (3)$$
where $C$ is the total coverage of community ($C = \sum C_i$); $i = 1$, tree layer; 2, shrub layer; 3, herb
layer, and the meaning of $i$ same below; $h$ is the thickness of the leaf layer for various growth types
($h = \sum h_i$), $W_i$ is the weighted parameter of diversity index of $i^{th}$ growth type, $C_i$ is the coverage of the
$i^{th}$ growth type and $h_i$ is the average thickness of the leaf layer of the $i^{th}$ growth type. Among different
growth type, the thickness of tree leaf layer is calculated at 33.3% the height of the tree layer, the shrub
layer is at 50% and the herb layer is at 100%.
The total diversity index of the community was calculated according to the following formula:
$$A = \sum W_i A_i \qquad (4)$$
where $W$ is the weighted parameters of the tree layer, shrub layer and herb layer. $A$ is the diversity
index of the tree layer, shrub layer and herb layer, which can be calculated using the formulae listed
below.
Species diversity indices were determined (Liu et al., 1997) as Shannon–Wiener's index of
diversity
$$H = -\sum_{i=1}^{s}(P_i \ln P_i) \qquad (5)$$
and Simpson's index of dominance was calculated as
$$D = 1 - \sum_{i=1}^{s} P_i^2 \qquad (6)$$
and Pielou's index of evenness was calculated as
$$J_{sw} = H/(\ln(S)) \qquad (7)$$
Finally, Patrick's index of richness was calculated as
$$R = S \qquad (8)$$

where $P_i$ is the relative important value of species $i$, and $S$ is the total number of species in the $i^{th}$ site.

Within each gradient, vegetation community, soil moisture and soil properties of the five sites were calculated as mean ± standard error (SE) of the mean. To depict the vertical structure of soil moisture, soil water content was divided into three layers: 0-30 cm soil moisture (SWC1), 30-100 cm soil moisture (SWC2), and 100-200 cm soil moisture (SWC3) in accordance to the fine roots distribution of herbs, trees and shrubs in this area (Fu et al., 2014). We averaged the soil moisture at each corresponding finer increment to obtain the value of SWC1, SWC2 and SWC3. Soil chemical properties, however, were analyzed using the mean values of 0-100cm due to the minor vertical variation. The annual average value and annual variability were used to depict the temporal variation of community characteristics and environment factors. The annual average of NDVI (NDVI_a), groundwater (GWT_a), 2cm soil moisture (SWC2cm_a), 100cm soil moisture (SWC100cm_a) were calculated by the mean values from 2000-2014. The annual variability of NDVI (NDVI_c), groundwater (GWT_c), 2cm soil moisture (SWC2cm_c), 100cm soil moisture (SWC100cm_c) were calculated by the mean values of change rate at each year.

Regression analysis was used to examine variation pattern. Exponential and polynomial regressions were fit to the data to best explain the statistical relationship. Pearson correlation was used to determine the strength of possible relationship between community characteristics and environmental factors. Significant differences were evaluated at the 0.05 and 0.01 level. Statistical analysis was performed using SPSS (ver. 18.0).

To depict the variation of desert riparian forests composition, we used Two-way Indicator Species Analysis (TWINSPAN, in WinTWINSPAN, version 2.3), a method of community hierarchical classification based on the importance value of each species (Hill, 1979), to classify the possible desert riparian forests community types. The importance value data for all plant species, obtained from the vegetation survey were used in this analysis and the cutoff levels of importance value for each class were set as: 0, 0.1, 0.2, 0.4, 0.6 and 0.9. To further separating the key influencing factors of the 18 environment variables, marginal and conditional effects of various variables were calculated through the Monte Carlo forward selection in RDA (Redundancy Analysis), which directly showed the significance and contribution rate of each factor. Marginal effects reflected the effects of the environmental variable on the community characteristics, while conditional effects reflected the effects

of the environmental variables on the community characteristics after the anterior variable was eliminated by the forward selection method. Since the redundant variables were eliminated and a group of key environmental factors was determined through the forward selection, this method allowed key variables to be determined through the strength of their effects and significance. Variation of community characteristics explained by the different group of environmental factors was analyzed using variation partitioning analysis. The significance of the resulting ordination was evaluated by 499 Monte Carlo permutations (Zhang and Dong, 2010). The Monte Carlo test and variation partitioning analysis were performed by the software program CANOCO (ver. 5.0) (Microcomputer Power, USA) (Braak et al., 2012).

## 3 Results

### 3.1 Vegetation community types and temporal changes of vegetation composition

Species composition at each site in the downstream Heihe River Basin is shown in Table S1 and the following five plant community types distributed across the 3000 m transect from river channel were obtained based on TWINSPAN classification (Fig. 2):

(i) Community I was an association of (Ass.) *Populus euphratica–Tamarix ramosissima* + herbs, found at sites 1, 2, 3, 4, 6, 7, 15 and 21. Although this community, with multiple layers of tree-shrub-herb, was typical at desert riparian forest, its coverage was relatively low (38.05%). The community was dominated by tree species *Populus euphratica* with sparse understory vegetation. *Tamarix ramosissima* was the only species of shrub layer and the herb layer was dominated by *Sophora alopecuroides*. This community mainly distributed near the river bank, mostly within 500 m from the river channel.

(ii) Community II was Ass. *Tamarix ramosissima–Lycium ruthenicum* + herbs, found at sites 5, 10 and 26. This community was constituted of shrub and herb layers with high community coverage of 81.43%. *Tamarix ramosissima* was the dominant species of the shrub layer with the importance value of 0.84-1.00. The herb layer contains both hygrophyte and xerophyte species, such as *Kochia scoparia* and *Peganum harmala*. This community was mainly distributed near the river bank (about 1000 m from the river channel).

(iii) Community III was *Tamarix ramosissima*, found at sites 8, 9, 20, 23 and 25. This community was mainly constituted of shrub layers, except that sparsely grown herbs existed at site 8. The community

was dominated by *Tamarix ramosissima* with average community coverage of 75.93% and mainly

distributed at the distance between 1000m – and 2000 m from the river channel.

(iv) Community IV was Ass. *Lycium ruthenicum–Tamarix ramosissima* + xerophytes herbs, found at

sites 12, 13, 14, 17, 18, 22, 24, 27, 32 and 34. This community mainly composed of shrub and herb

layers with average community coverage of 68.86%. *Lycium ruthenicum* was the dominant species of

the shrub layer (importance value = 0.42-0.77), while the dominant xerophytic herb species were

*Sophora alopecuroides* and *Suaeda salsa*. It was mainly distributed between 1500m and 2500 m from

the river channel.

(v) Community V was Ass. *Tamarix ramosissima–Lycium ruthenicum–Reaumuria songarica*, found at

sites 11, 16, 19, 28, 29, 30, 31, 33 and 35. This community was the transition community from desert

riparian shrub forests to desert shrub community, indicated by the presence of *Reaumuria songarica*, a

typical desert shrub. *Tamarix ramosissima* was the dominant species of the shrub layer and mainly exist

in the form of shrub dune, with the importance value of 0.38-0.93. The *Karilinia caspica and*

*Phragmites communis* herbs only existed in one sampling site and they were only sparsely distributed.

This community was mainly distributed around 2500-3000 m from the river channel, with a relatively

low community coverage (54.40%).

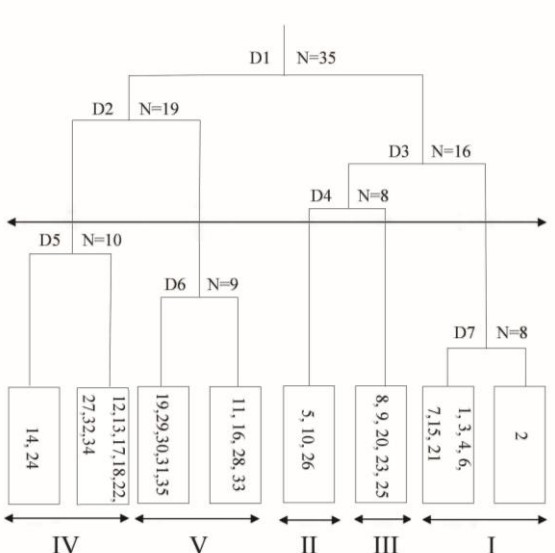

**Figure 2.** The dendrogram of the sampling sites based on the TWINSPAN classification

Note: Number 1-35 represents the site number of the sampling sites. D is for the classification levels and N is for the numbers of

sampling sites for the classification. I to V represent community I to V. Arrows depicted all the sites were divided into five major

groups after the fourth classification.

Vegetation composition change in each community type (I to V) was obtained from the land use map
from 2000 to 2014 (Fig. 3). Among five community types, community V underwent most changes, with
22.22% sites change from sparse forest to grassland, 22.22% from grassland to shrubland and 22.22%
from bareland to grassland, respectively. The majority (>60%) of vegetation composition remain
unchanged in community I to IV, with the following exceptions: (i) 37.5% sites in community I
changed from shrubland to sparse forest and from bareland to grassland, (ii) 33% and 20% sites in
community II and III changed from bareland to grassland and from sparse forest to grassland,
respectively. and (iii) 20% sites in community IV changed from sparse forest to grassland and another
20% from grassland to shrubland (Fig. 3).

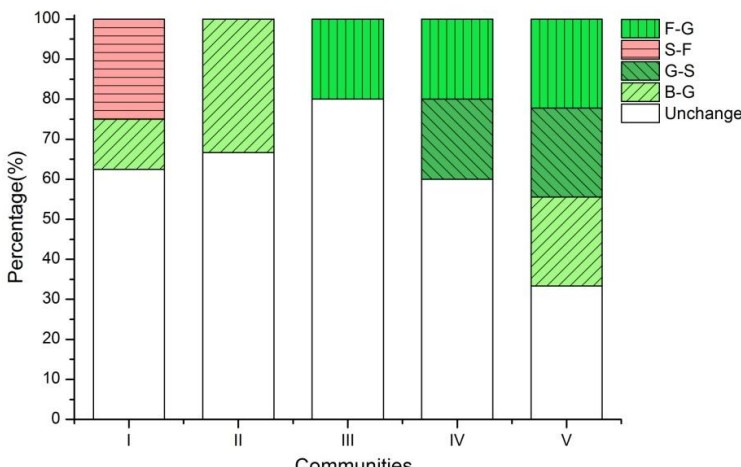

**Figure 3.** The percentage changes of vegetation composition in each community from 2000-2014
F-G: change from sparse forest to grassland; S-F: change from shrubland to sparse forest; G-S: change from
grassland to shrubland; B-G: change from bareland to grassland.
**3.2 The spatial and temporal variation of community characteristics in desert riparian forest**
Community characteristics formed different patterns along the distance from the river channel (Fig. 4).
Vegetation community height and density dropped rapidly after 500 m (Fig. 4a, b), while community
coverage formed a bimodal pattern, peaked at the distance of 500-1000 m and 3000 m, respectively
(Fig. 4c). The variation of vertical structure was depicted by the following hierarchical coverage (Fig.
4d): (i) the tree layer mainly existed within 1000 m (ii) the shrub peaked around 1500-2000 m, and (iii)
the herb fluctuated along the distance gradient, peaking at 500 m and 2500-3000 m from the river
channel. All diversity indices showed a bimodal pattern along the distance from river channel. The
Shannon-Wiener diversity index, Pielou evenness index and Patrick richness index peaked at 1000 m
and 3000 m (Fig. 4e-g). The Simpson dominance index, however, formed an opposing trend to the
other three diversity indices, by peaking at 500 m and 2000 m where the other indices were at their low
level (Fig. 4h).

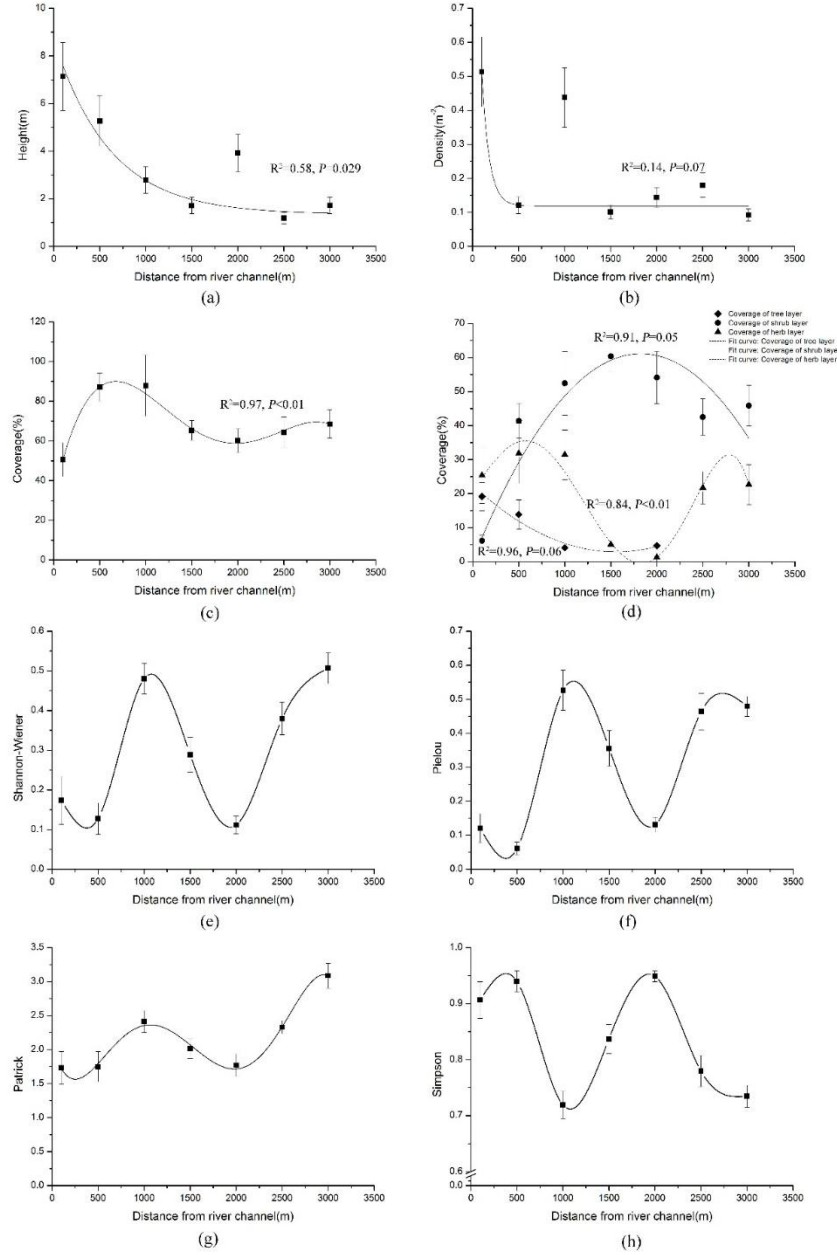

**Figure 4.** The variation of community structure and diversity along the distance from the river channel.
The temporal variation of community characteristics was depicted by the variation of NDVI (Fig.
5). At different gradients, temporal variation of NDVI showed similar pattern with an overall increasing
trend throughout the research period except a little decrease during the initial years (2000-2002). NDVI
decreased along the distance from the river channel, with the highest and the lowest NDVI values were
found closest (100 m, 500 m) and furthest away from the river channel (3000 m), respectively (Fig.5a).
NDVI annual variability, however, showed a contrary trend, increasing as it moved away from the river
channel, but decreasing as it moved closer to the river channel (Fig. 5b).

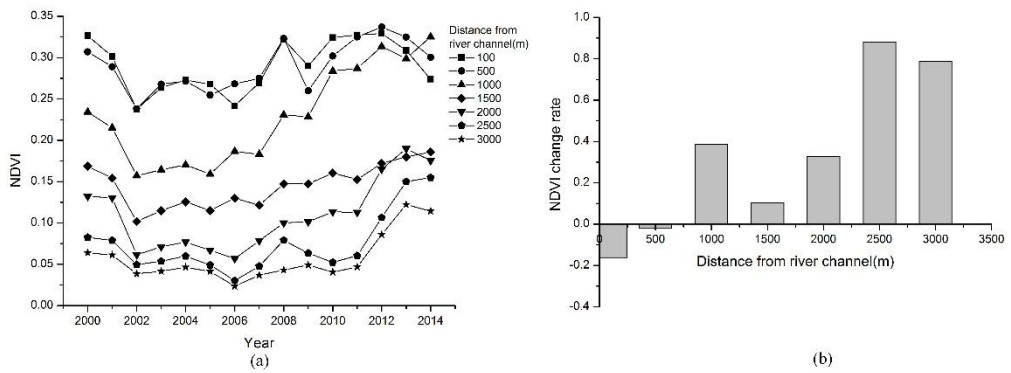

                                    (a)                                  (b)

**Figure 5.** The variation of NDVI (a) and annual variability of NDVI (b) from2000 to 2014 at different distance

7                                          from the river channel.

**3.3 The spatial and temporal variation of water availability and soil properties**
Water availability and soil properties varied significantly along the distance from the river channel (Fig.
6). SWC1 (0-30 cm soil moisture) and SWC2 (30-100 cm soil moisture) peaked at the distance of
500-1000 m and 2500 m, following the same pattern with vegetation community coverage, and
diversity indices (Fig. 4 c-f). SWC3 (100-200 cm soil moisture), however, showed a different pattern
by peaking at the distance of 1000 m from river channel and dropped rapidly after 2500 m (Fig. 6 a).
The proportion of silt and clay was highest at the distance of 1000 m from the river channel (Fig. 6 c),
while bulk density reached its lowest point (1.07 g·cm$^{-3}$) (Fig. 6 b). The variation of SOM, TN, TP
showed the similar pattern with vegetation diversity along the gradient (Fig. 4 e-g, Fig. 6 d-g). They
generally decreased along the distance from river channel and reached a relatively high value at the
distances of 500 m and 2000-2500 m. The total salt content peaked at the distance of 1000 m (2.57%)
and dropped gradually until the end of the gradient.

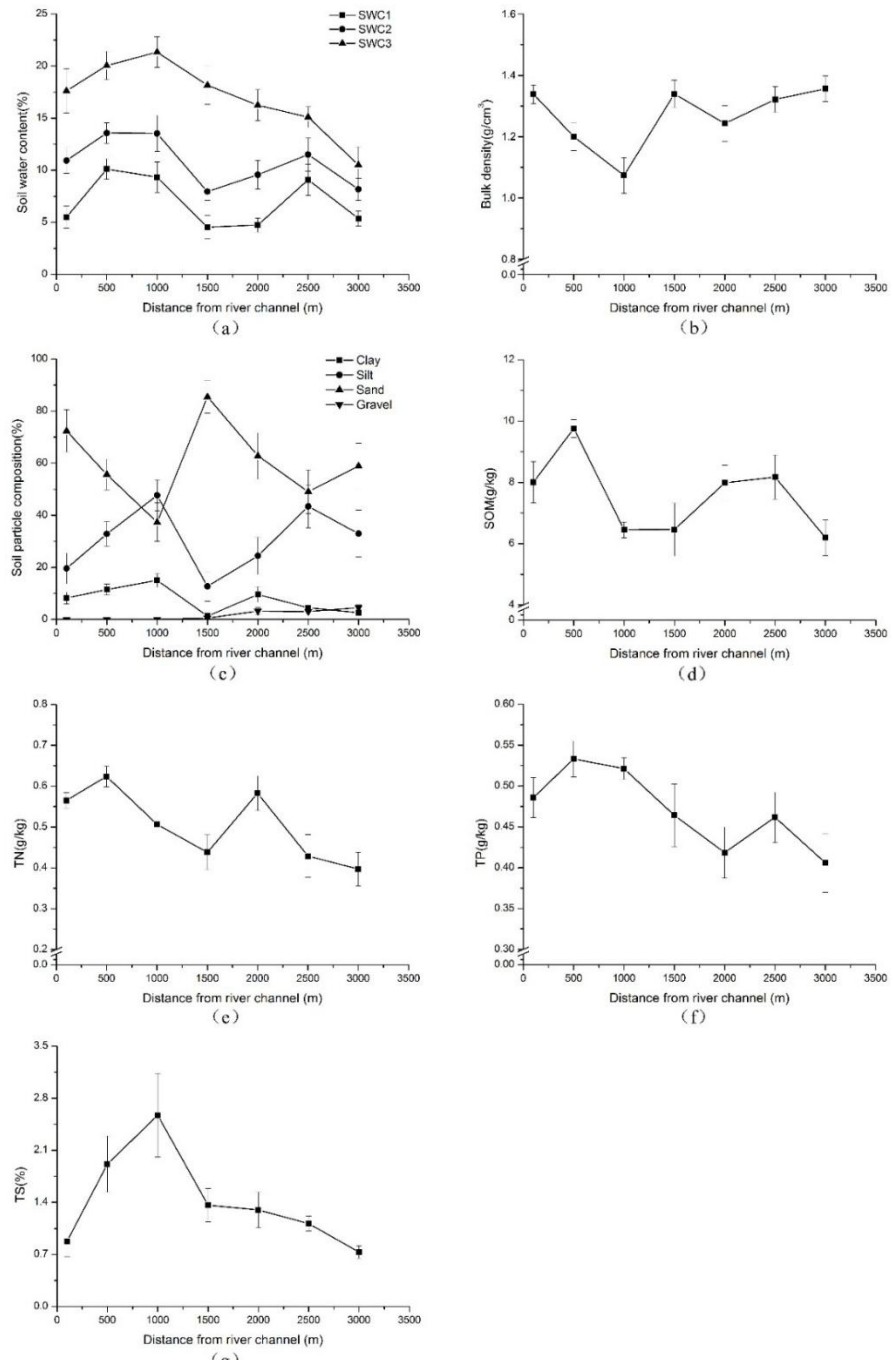

**Figure 6.** The variation of soil moisture (a), soil bulk density (b), soil particle composition (c), soil organic matter (d), total nitrogen (e), total phosphorus (f), total salinity (g) along the distance from river channel.

SWC1, 0-30cm soil moisture; SWC2, 30-100cm soil moisture; SWC3, 100-200cm soil moisture; BD, bulk density; SOM, soil organic matter; TN, total nitrogen; TP, total phosphorus; TS, total salt content.

The temporal variation of water availability and soil properties was depicted by soil moisture, groundwater table and runoff. Soil moisture decreased with the distance from river channel (Figs.7 a, b) and different gradient formed a similar temporal variation pattern. Shallow (2 cm) soil moisture showed

greater fluctuation than deep (100 cm) soil moisture, which was almost constant with time. The depth
of groundwater table increased consistently across different gradients since 2000, following the
downstream runoff which doubled during the research period, from $6.5 \times 10^8$ m$^3$ in 2000 to $13 \times 10^8$ m$^3$ in

4 2014.

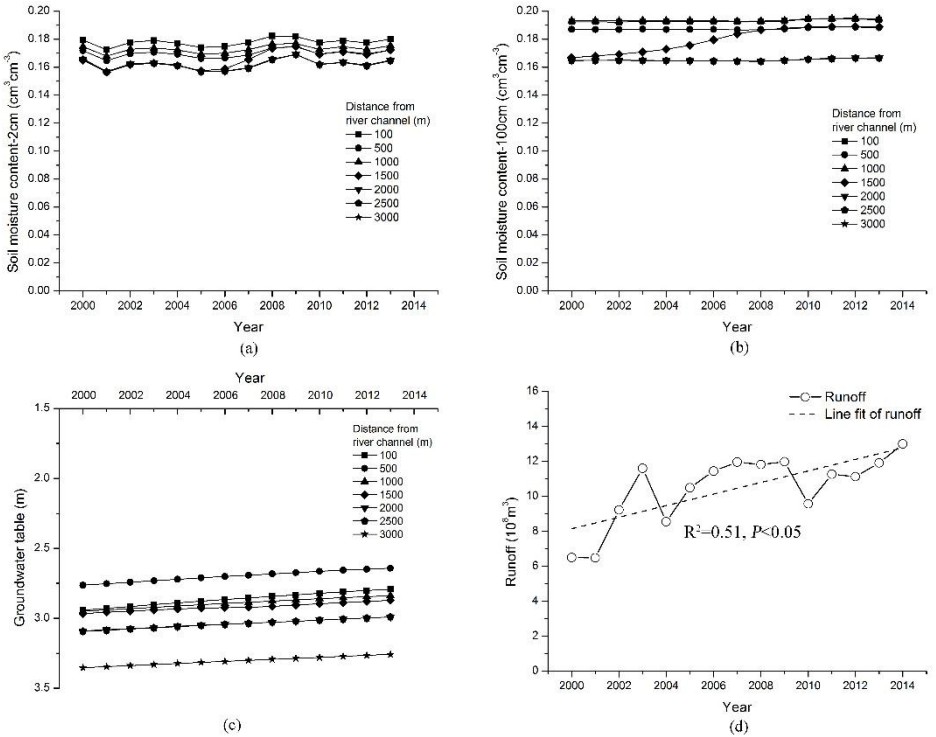

**Figure 7.** The variation of 2 cm soil moisture (a), 100 cm soil moisture (b), groundwater table (c), runoff (d) from
2000-2014 at different distance from the river channel.
**3.4 Pearson correlation between community characteristics and environmental factors**
Pearson correlation analysis between community characteristics and environmental factors is shown in
Table 1. The community density showed significant positive correlation with SWC2, SWC3,
SWC2cm_a and SWC100cm_a, but negative correlations with BD and GWT_a. Community coverage
positively correlated with all the three layers of soil moisture ($P<0.0.1$) but negatively correlated with
BD. Tree and shrub layers layer influenced by GWT_a and BD, respectively, while herb layer
positively correlated with SWC1 and SCW3. Among the diversity indices, the Patrick richness index
was significantly correlated with SOM and gravel, while Simpson domination index was significantly
correlated with sand (r= 0.354, $P<0.05$) and silt (r=$-$0.344, $P<0.05$). As for temporal variation of
community characteristics, NDVI_a was mainly influenced by soil moisture (SWC1, SWC2, SWC3),

soil particle composition (clay, gravel) and bulk density, while NDVI_c was significantly correlated with SWC3, gravel and TS.

With runoff as the main water resource in the downstream Heihe, there was time lag between the increase of runoff and NDVI. The correlation coefficient between NDVI and runoff was measured to examine the relationship between runoff and the same year's NDVI, while correlation coefficient between one year lag NDVI and runoff was measured to exam the relationship between runoff and the next year's NDVI. One year lag NDVI-runoff correlation coefficient decreased significantly with the distance from river channel (P=0.086), as opposed to insignificant variation of NDVI-runoff correlation coefficient along the distance from river channel (Fig. 8).

**Table 1.** Pearson correlation between community characteristics and environmental factors

| | H | R | C | Jsw | Height | Density | Cover-a | Cover-t | Cover-s | Cover-h | NDVI_a | NDVI_c |
|---|---|---|---|---|---|---|---|---|---|---|---|---|
| SWC1 | 0.255 | 0.167 | -0.286 | 0.182 | -0.088 | 0.251 | **0.545** | -0.017 | 0.168 | **0.514** | **0.430** | 0.188 |
| SWC2 | 0.046 | -0.072 | -0.098 | 0.067 | -0.114 | **0.382** | **0.439** | 0.007 | 0.280 | 0.263 | **0.469** | 0.254 |
| SWC3 | 0.142 | 0.157 | -0.147 | 0.111 | -0.242 | **0.362** | **0.448** | -0.142 | 0.175 | **0.382** | **0.445** | **0.506** |
| Clay | 0.112 | 0.005 | -0.128 | 0.045 | 0.048 | 0.290 | 0.204 | 0.037 | -0.093 | 0.272 | **0.398** | 0.125 |
| Silt | 0.308 | 0.117 | **-0.344** | 0.311 | -0.121 | 0.111 | 0.321 | -0.071 | 0.247 | 0.168 | 0.185 | -0.115 |
| Sand | -0.327 | -0.148 | **0.354** | -0.306 | 0.130 | -0.165 | -0.307 | 0.076 | -0.166 | -0.217 | -0.212 | 0.125 |
| Gravel | 0.226 | **0.350** | -0.155 | 0.179 | -0.284 | -0.081 | -0.185 | -0.173 | -0.179 | 0.011 | **-0.413** | **-0.396** |
| BD | 0.174 | 0.282 | -0.127 | 0.123 | -0.041 | **-0.353** | **-0.350** | 0.049 | **-0.465** | 0.063 | **-0.354** | -0.050 |
| SOM | -0.256 | **-0.398** | 0.187 | -0.102 | 0.193 | 0.058 | -0.192 | 0.116 | -0.121 | -0.296 | -0.025 | -0.009 |
| TN | -0.191 | -0.333 | 0.138 | -0.060 | 0.101 | 0.032 | -0.278 | 0.112 | -0.296 | -0.223 | -0.006 | 0.108 |
| TP | -0.238 | -0.303 | 0.198 | -0.098 | 0.116 | 0.022 | -0.181 | 0.084 | -0.090 | -0.288 | -0.018 | 0.194 |
| TS | -0.139 | -0.125 | 0.111 | -0.099 | -0.184 | 0.271 | 0.011 | -0.086 | 0.034 | -0.131 | -0.140 | **0.382** |
| GWT_c | 0.094 | -0.028 | -0.133 | 0.228 | -0.074 | 0.001 | -0.137 | 0.102 | -0.060 | -0.189 | -0.286 | 0.040 |
| SWC2cm_c | 0.113 | 0.085 | -0.117 | 0.084 | -0.161 | 0.098 | -0.027 | -0.093 | -0.029 | -0.024 | -0.177 | 0.119 |
| SWC100cm_c | 0.171 | 0.185 | -0.165 | 0.109 | -0.116 | -0.080 | 0.073 | -0.096 | 0.107 | 0.038 | -0.198 | 0.141 |
| GWT_a | -0.022 | -0.226 | -0.050 | 0.127 | 0.300 | **-0.343** | -0.092 | **0.352** | 0.017 | -0.131 | 0.042 | 0.004 |
| SWC2cm_a | -0.169 | -0.270 | 0.129 | -0.096 | 0.013 | **0.405** | -0.184 | 0.103 | -0.224 | -0.183 | 0.160 | 0.144 |
| SWC100cm_a | -0.085 | -0.194 | 0.047 | -0.014 | -0.094 | **0.403** | -0.137 | -0.046 | -0.206 | -0.150 | 0.090 | 0.140 |

Significant correlations (P<0.05) are shown in bold and significant correlations (P<0.01) in bold with underline.

R, Patrick richness index; $J_{sw}$, Pielou evenness index; H, Shannon–Wiener diversity index; C, Simpson domination index; a, total plant community; t, tree layer; s, shrub layer; h, herb layer; NDVI_a, annual average of NDVI; NDVI_c, average annual variability of NDVI; SWC1, 0-30cm soil moisture; SWC2, 30-100cm soil moisture; SWC3, 100-200cm soil moisture; BD, bulk density; SOM, soil organic matter; TN, total nitrogen; TP, total phosphorus; TS, total salt content. 0-20cm soil particle composition were analyzed in the laboratory for the silt (<0.02mm), clay (0.02-0.05 mm), sand (0.05-2 mm), and gravel (>2mm) contents by using Mastersizer 2000. Soil chemical properties at 0-20, 20-40, 40-60, 60-80 and 80-100 cm and the average value of 0-100cm were used in the analysis. GWT_a, annual average of groundwater table; SWC2cm_a, annual average of 2cm soil moisture; SWC100cm_a, annual average of 100cm soil moisture; GWT_c, annual variability of groundwater table; SWC2cm_c, annual variability of 2cm soil moisture; SWC100cm_c, annual variability of 100cm soil moisture;

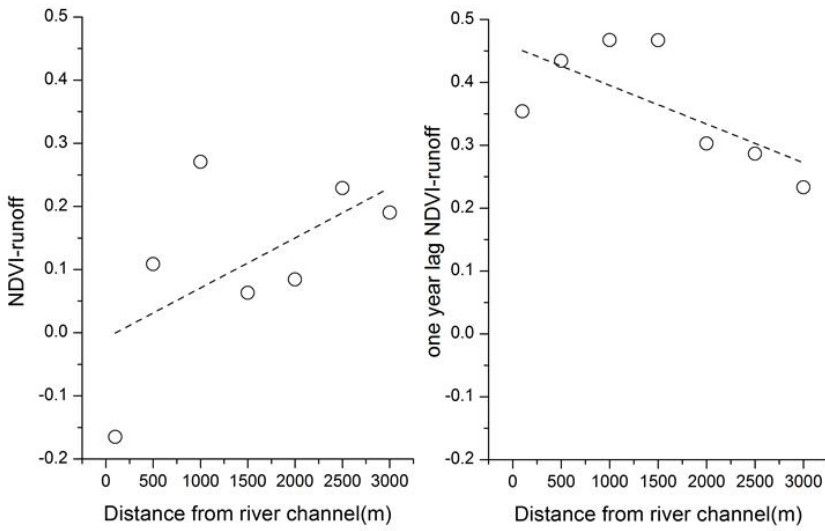

**Figure 8.** Pearson correlate coefficient of NDVI-runoff and one year lag NDVI-runoff at different distance from
river channel.
**3.5 Key environmental factors that influenced community characteristics**
To further examine the key environmental factors that controlled the variation of vegetation indices (e.g.
community diversity, structure, NDVI), redundant variables were eliminated by a forward selection
method. Table 2 shows the key influencing factors based on the marginal and conditional effects of 18
variables under the Monte Carlo test in the process of forward selection. All the environmental factors
explained 74% variance of total variance. In the Monte Carlo test of forward selection ($P<0.05$), SWC1,
SWC3, BD and SWC100cm_a were regarded as the key environmental factors influencing the
variation of community characteristics. A total 71.62% of the environmental information was extracted
by the key environmental factors, and SWC1 contributed the most information (27.03%). To further
investigate the variation explained by spatial heterogeneity factors and temporal variation factors, we
divided those 18 factors into two groups for partitioning analysis (Table 3). Spatial heterogeneity
factors explained 43.5% vegetation variance and accounted for 98.4% of the total variance explanation,
while temporal variation factors only explained 15.9% vegetation variance, accounting for 35.9% of
total variance explanation. These two groups of factors jointly explained 15.2% vegetation variance,
accounting for 34.3% the total variance explanation.
**Table 2.** The selection of the key influencing factors based on the marginal and conditional effects obtained from
the forward selection of Monte Carlo test.

| Environmental factors | Marginal effects Percentage of variance explained (%) | Environmental factors | Conditional effects Percentage of variance explained (%) | P value | R value (%) |
|---|---|---|---|---|---|
| SWC1 | 20.2 | SWC1 | 20 | 0.002 | 27.03 |
| SWC3 | 18.8 | SWC3 | 14 | 0.004 | 18.92 |
| SWC2 | 12.3 | BD | 10 | 0.006 | 13.51 |
| BD | 11.4 | SWC100cm_a | 9 | 0.018 | 12.16 |
| TN | 7.1 | GWT_a | 4 | 0.078 | —— |
| Silt | 7 | GWT_c | 3 | 0.096 | —— |
| Sand | 6.1 | TP | 2 | 0.25 | —— |
| SOM | 4.1 | Clay | 2 | 0.282 | —— |
| Clay | 3.8 | TN | 2 | 0.296 | —— |
| SWC2cm_a | 3.7 | SWC2cm_a | 2 | 0.308 | —— |
| TP | 3.6 | SWC100cm_c | 1 | 0.444 | —— |
| Gravel | 2.6 | SWC2cm_c | 3 | 0.112 | —— |
| SWC100cm_a | 2.5 | SWC2 | 1 | 0.62 | —— |
| GWT_c | 1.8 | Silt | 1 | 0.636 | —— |
| GWT_a | 1.4 | TS | <0.1 | 0.788 | —— |
| SWC100cm_c | 0.6 | SOM | <0.1 | 0.932 | —— |
| TS | 0.5 | Sand | <0.1 | 0.992 | —— |
| SWC2cm_c | 0.1 | Gravel | <0.1 | 0.96 | —— |
| | | Total | 74 | 0.036 | —— |

$R$ value represents the relative proportion of individual explanation to the total variance explanation.
**Table 3.** The percentage of community characteristic variations explained by key environmental factors.

| Fraction | Variation | % of All | % of Explained | F | P |
|---|---|---|---|---|---|
| a | 0.43539 | 43.5 | 98.4 | 5.9 | 0.008 |
| b | 0.1588 | 15.9 | 35.9 | 4 | 0.088 |
| a+b | 0.1519 | 15.2 | 34.3 | 2.2 | 0.016 |
| Total Explained | 0.44229 | 44.2 | 100 | 5.9 | — |

a: spatial distribution factors, including SWC1, SWC2, SWC3, BD, clay, silt, sand, gravel, SOM, TN, TP, TS; b: temporal
factors, including SWC2cm_a, SWC100cm_a, GWT_a , SWC2cm_c, SWC100cm_c, GWT_c; Variation: the variance explained
by different fraction when the total variance is 1; % of All: the proportion of variation explained by different fraction; % of
Explained: the relative proportion of individual explanation to the total explanation;
**4 Discussion**
**4.1 The distribution pattern and temporal variation of community characteristics in desert**
**riparian forest**
The characteristics and indices of desert riparian forests formed different patterns along the distance
from the river channel in the downstream Heihe River Basin. Community height and density declined
significantly as dominant species changed from trees to riparian-desert shrubs along the distance
gradient. The community coverage reached local maxima at the distance of 1000 m and 3000 m where
community consisted of diverse shrub and herb layers. Our findings were different from those in
relatively humid region (e.g., coastal region or boreal forest), which suggested that riparian forest
species diversity either decreased or formed a unimodal pattern with increasing distance from the
stream (Pabst and Spies, 2011; Macdonald et al., 2014). These variation patterns of community
diversity can illustrate how community response to the ecological gradient (Zhu et al., 2013) and
interact with environmental factors in this resource limited region (Oksanen and Minchin, 2002).
Although located quite far from the river, soil moisture (e.g. SWC1, SWC2, and SWC3) reached its
maximum at 1000 m from river channel (Fig. 6), supporting rich vegetation community (multiple
layers of tree-shrub-herb). High soil moisture (up to 100 cm deep) provided adequate water resource
for the growth of diverse species as soil moisture also explained for 49.95% of vegetation variance.
In addition, the presence of deep-rooted tree, *Populus euphratica* could benefit the growth of
shallow-rooted species (e.g. herbs) by redistributing the deep soil water to the shallow layer as a
strategy of mutualism (Hao et al., 2013). At further distance from the river (3000 m), high species
diversity could be supported by the presence of fine soil particles, which resulted in relatively high soil
infiltration capacity and soil nutrition around the shrub patches ('fertile islands') (Ravi et al., 2010;
Zhou et al., 2015). Although situated at the transition region (from riparian forest to desert shrubs), the
soil here was still rich in fine particles (clay and silt; 35.6%), brought by the interaction between wind
erosion and shrubs (Ravi et al., 2009). These 'fertile islands' allowed the growth of some xerophytic
herbs, increasing the level of diversity in this gradient (Stavi et al., 2008; Ravolainen et al., 2013). By
contrast, Simpson dominance index peaked at the distance of 500 m and 2000 m where other indices
were at their low level (Fig. 4 h). We suggested that inter-species competition for water and nutrient
resources could be responsible for the trend (Maestre et al., 2006; Boever et al., 2015). The dominant
species with high important value (i.e., trees and shrubs at 500 m and 2000 m, respectively) often had
high competition for resources, halting the growth of other species (i.e., herbs) (Koerselman and
Meuleman, 1996). In these sites, low number of species indicated low community diversity and the
dominant species made a large contribution to the diversity index of the community (Zhu et al., 2013),
resulting in a large domination index (Fig. 4 h).

25       Since the implementation of ecological water conveyance project in 2000, the vegetation in desert

riparian forest has recovered significantly, shown by the increasing NDVI at different distances from
the river channel (Fig. 5 a). Although there was initial decrease of NDVI during 2000-2001, likely due
to the one year lag effect of runoff and relatively low runoff at these early years (Jin et al., 2008; Ge et
al., 2009), NDVI generally increased with the restoration time. The conversion of low coverage
community (e.g. sparse forest land, bareland land) to shrubland and grassland at the distance of
2000-3000 m from the river channel likely contributed to the increase in NDVI with better water
availability in Heihe River Basin (e.g. increase of surface soil moisture and elevate of groundwater). In
contrast, NDVI around the river bank underwent a slight degradation in the recent years (2012-2014),
likely result from the conversion of shrubland to sparse forest land (Fig. 5 b). In the arid zone, grazing
is mainly limited to the region near the river bank due to the abundance of palatable grass and available
of drinking water, which may hinder vegetation recovery compared to other gradients (Todd., 2006). In
addition, high soil moisture and low salinity supported the regeneration of *Populus euphratica* trees. As
they became the dominant species, they limited the growth of other species due to inter-species
competition, leading to decrease in NDVI. High tourism pressure may also hinder vegetation growth
during the growing season (May to October) since *Populus euphratica* trees are becoming popular
tourist destination (Hochmuth., 2014).
**4.2 Factors influencing the distribution pattern and temporal variation of desert riparian forest**
Among the environmental factors, changes in water availability associated with soil properties are
considered as the most important selective forces shaping ecosystem stability in hyperarid zone
(Rosenthal and Donovan, 2005; Ravi et al., 2010; Feng et al., 2015). Our study showed that
environmental factors explained 74.0% vegetation variance in total (Table 2), which indicated that both
spatial heterogeneity and temporal factors play important role in determining the community
characteristics of desert riparian forests in the Heihe River Basin.
Among those factors, SWC1, SWC3, BD, annual average of 100 cm soil moisture were
considered the key influencing factors, with SWC1 and SWC3 contributed to 45.95% of the total
explanation of vegetation variance. SWC1 (0-30 cm soil moisture) contributed to high coverage of herb
layers as it become the main water source for the dominant herb species, such as *S. alopecuriodes* and
*K. caspica* whose fine roots mainly distributed within 30 cm from the surface soil (Fu et al., 2014).
SWC2 and SWC3 mainly influenced the community density and the annual fluctuation of NDVI.
SWC2 (30-100 cm soil moisture) was the main water resource for shrubs such as, *T. ramosissima*.
SWC3, recharged by flood-raised groundwater table (Liu et al., 2015), was the water source for
phreatophyte like *P. euphratica* or desert shrubs (Yi et al., 2012). As tree and shrub contributed greatly
to the community composition, the increase in SWC2 and SWC3 could significantly promote the
vegetation growth, increasing the community density and NDVI. All three layers of soil moisture

positively affected both community coverage and the annual average of NDVI (NDVI_a), which indicated that improved water availability directly promoted vegetation recovery in different gradients and high community coverage in this current stage.

Among spatial heterogeneity factors, soil physical properties were also important in determining vegetation community with BD accounted for 13.51% of the total explanation of vegetation variance. Bulk density mainly influenced community density, community coverage, shrub coverage, and annual average of NDVI, while soil composition (clay, silt, gravel) mainly affected the Simpson diversity indices, annual average of NDVI, and annual average change of NDVI. Bulk density and soil composition are critical for water and nutrient holding capacity and the ability of absorbing soil nutrition (Stirzaker et al., 1996; Meskinivishkaee et al., 2014). Soil with low bulk density is characterized by high porosity, which allows more water to infiltrate into the deep soil, promoting the growth of deep root vegetation and benefiting community density, coverage and annual average NDVI in each gradient. While, soil with high bulk density often consisted of high silt and sand, but low percentage of clay which resulted in low water holding capacity in the surface soil (Ravi et al., 2010) and possibly inducing the drought stress to the vegetation community (Stirzaker et al., 1996). Such process constrained vegetation recovery especially herbs, which contributed greatly to the community coverage, density and diversity, It also hindered the NDVI increase, resulting in low diversity and a large domination index of the community. Soil nutrition explained no more than 3% of vegetation variance, and we found that SOM negatively correlated with species richness. This finding was different from the commonly positive relationship between SOM and species richness in semiarid zone (e.g. Loess Plateau) found in previous study (Jiao et al., 2011; Yang et al., 2014). Although SOM content determined soil nutrient storage and supply of available nutrients, our sites in hyperarid zone were often characterized by barren soil with less than 1% soil organic matter (Fig. 5d). Such low amount of SOM might not be able to boost the growth of various species in desert riparian forests (Wang et al., 2016). At the same time, the dominant species (i.e., *P. euphratica* and *T. ramosissima*), despite producing high amount of litter, they also had high competition for resources, thus halting the diversity and growth of other species (Su, 2003).

The temporal variation factors partly explained vegetation variations (35.9%) and SWC100cm_a was considered the key influencing factor. Along with GWT_a and SWC 2cm_a, they contributed to the recovery of desert riparian forest, shown by the increase in community density (Table 1). As soil

moisture up to 100 cm deep is beyond the impact of increasing evaporation under climate change
(Zhang et al., 2015a) and less influenced by the fluctuation of groundwater, it could be considered as
reliable water source for vegetation. We, however, found that the coverage of tree layer showed a
negative relationship with GWT_a, contrary to studies in Tarim river where a sharply decrease of
groundwater level was observed along the distance from river channel (Chen et al., 2015). In Heihe
River, the groundwater did not fluctuate much and the perennial groundwater table remained above 4 m.
High water table allowed the deep-rooted trees to face more competition from shallow-rooted species
and therefore with the deepening of groundwater, the habitat become more suitable for the growth of
deep-rooted vegetation (e.g. tree and shrub) than the shallow-root one (e.g. herb) (Ditommaso et al.,
1989). Compared to the annual average of water availability in each gradient, the annual variability of
soil moisture and groundwater due to runoff did not have significant impact on community
characteristics, most likely because the latter did not fluctuate much during 2000-2014 (Fig.7).
Ecohydrological processes in riparian zone such as seepage, interflow, groundwater movement and
vegetation evapotranspiration (Liu et al., 2012) lifted up groundwater and soil water condition,
moderating the effect of rapid runoff increase. The recovery of vegetation was therefore more likely
benefited from long term improvement in water condition instead of the annual water variability (i.e.,
runoff) as mutual effect of the aforementioned ecohydrological processes would result in a more stable
re-charge of soil moisture and groundwater.
**4.3 Community resilience of desert riparian forests and implications for ecological protection**
As the main communities in the downstream Heihe River Basin, desert riparian forest strongly
influenced the ecosystem resilience and resistance against disturbance. Studies have shown that
species-rich communities can maintain ecosystem functions during stress-based perturbations due to
the complementary of function traits and ecological redundancy (Luck et al., 2013; Isbell et al., 2015).
Although community diversity was generally low in the downstream Heihe River Basin at most
gradients, it was significantly higher at 1000 m and 3000 m gradients (Fig 4). High resistance to
drought stress was observed at these gradients, with trees and shrubs lifting up water from the deep to
the shallower layer as a strategy of mutualism (Hao et al., 2013). Since trees and shrubs contributed
differently in the ecosystem functions (e.g. trees mainly contribute to carbon storage while shrub and

herb to sand fixation), they could maintain a stable habitat after drought stress and/or human disturbance (Cheng et al., 2007; Krieger et al., 2001; Lu et al., 2015). In contrast, communities at the other gradients could easily undergo degradations due to low resilience under disturbance (e.g., drought stress, grazing and tourism) such as those already happened at 500 m gradient, indicated by decreasing NDVI in these recent years (Fig.5a). Exposure to human disturbance, including trampling by livestock might potentially destroy the soil physical properties, reducing the ecosystem services such as water and soil conservation (Greenwood and Mckenzie, 2001; Zhao et al., 2012; Daryanto et al., 2013).

Our study showed that water availability and spatial heterogeneity of soil properties were the main driving forces for the spatial distribution and temporal variation of restored desert riparian forest at Heihe River Basin. Since the influence of ecological water conveyance was mainly limited to 1000 m distance from river (Si et al., 2005; Guo et al., 2009), projected rise in temperature could lead to the collapse of riparian vegetation (e.g. *Tamarix ramosissima*, *Lycium ruthenicum*) at further gradients, resulting in decrease of ecosystem service (e.g. sand fixation and carbon storage). In addition to potential threat posed by climate change, the periphery of the river is also more likely to be disturbed by grazing and heavy tourism pressure (Zenner, et al. 2012). Exposure to human disturbance, including trampling by livestock might potentially destroy the soil physical properties, reducing the ecosystem services such as water and soil conservation. To halt degradation in this critical zone, we suggested the development of natural channels that perpendicular to the river to fully extend the influence scope of ecological water conveyance and benefit the regions far from the river bank (Zhang et al., 2011b). At the same time, multiple conservation measures such as: (i) setting critical fence area for ecological protection, and (ii) constructing artificial shield or establishing straw checker boards on the bare land to prevent land degradation, are recommended around the periphery of the river.

**5 Conclusions**

Through extensive field observations at multiple desert riparian forests locations and analyses of long-term remote sensing images, we found that species diversity indices formed bimodal patterns instead of unimodal pattern. In locations with high diversity indices (1000 m and 3000 m), high community resilience was maintained by the multiple interactions between vegetation and soil properties. Still, these locations are facing challenge under climate change and intensive human disturbance.

Extending the distance of ecological water conveyance is therefore recommended to recharge the
surface soil moisture and benefit the growth of ground cover (i.e., herb species). Despite the increasing
NDVI trend, areas with low diversity (within 500 m from river channel) already underwent degradation
in recent years. Multiple conservation measures that protect the soil structure (e.g., artificial soil cover
and livestock grazing exclusion) are recommended for this region to reduce the adverse effects of
grazing on soil properties. Unless these necessary precautions are taken, desert riparian forests may
become restricted to the periphery of the river and experience significant community transition under
projected climate change scenario and more intensive human disturbance.
*Acknowledgements.* This work was supported by the National Natural Science Foundation of China
(No.91425301). Wenwu Zhao acknowledges the support from China Scholarship Council (No.
201406045031). We are also grateful to the Alashan Research Station of Cold and Arid Region
Environment and Engineering Research Institution, Chinese Academy of Sciences for their support and
contributions to the fieldwork. We are thankful for reviews from the editor, Prof. Kevin Bishop, and
two anonymous reviewers, which significantly improved the quality of this article.

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
