# Peer review of "The spatial distribution and temporal variation of desert riparian forests and their influencing factors in the downstream Heihe River Basin, China"

_Hydrology and Earth System Sciences, 2016_

## Referee Comment (RC1) · Anonymous Referee #1 · 16 Jun 2016

This work presents soil water content and biogeochemical data to explain how riparian vegetation changes as distance from the river increases. Vegetation is characterized by species composition and diversity, and occurrence and coverage of different plant functional types. The topic is overall relevant for readers of HESS. The manuscript is relatively clear, but might benefit from proofreading by a native English speaker. Despite the interest of the topic, I have some concerns regarding the analyses conducted and the mismatch between the ecological processes causing the observed vegetation patterns, and the one-time soil sampling adopted for this study.

Major issues

1) Ecological processes vs. one-time sampling. The plant communities examined in this work are the result of decade- if not century-long successional dynamics, but they are treated as if they are the result of short term processes. I refer specifically to soil water content, used as a predictor of vegetation community despite being measured only once. How representative are these water content measurements of the long-term water availability? Other soil properties vary at slower rates and could be more meaningful predictors (texture, SOM). How old are the trees and shrubs in this community? Are these communities shaped by the time they spent growing on a given soil (no information is provided to this regard), or by the edaphic properties of a given site (focus of the current study)? No data are reported on the variability in river discharge – how dynamic is the riparian environment? How frequent are flooding events that can re-shape the community (and soil properties)? Without this information, it is difficult to disentangle time effects from site effects.

2) Many of the measurements used as predictors are partly correlated, making it difficult to interpret the regression results. For example, soil water content is related to texture (as noted in P4, L18). Fine textured soils can hold more water, and this effect would appear in the gravimetric water content measurements. Total nutrient (TN and TP, which I assume include organic N and P) are also correlated to SOM, since large SOM stocks are associated with large N and P stocks (as noted in P20, L30). Due to these correlations, it seems difficult to apply regression approaches that assume independence, as in this case (if I interpreted the approach correctly).

3) The conclusions are based on too short-term a study to be really useful for planning. Either a long-term monitoring or a different study to identify possible historical reasons for the observed patterns would provide (or not!) support to a possibly large and expensive conservation project.

Minor issues

I am listing here only some of the small editorial issues in this MS – better to ask a native English speaker to give a thorough proofreading.

P2, L3: "focused" rather than "stressed" P2, L11: optimum in which sense? Is biomass higher around 1000 m, or what criteria was used to establish what the 'best' conditions are? P2, L19-20: it would be better to write if the mentioned influences are positive or negative. P3, L3: vague – what ecosystem services are important in this specific context? P3, L14: the term "ecological water conveyance" is not entirely clear? Is there a more commonly used term? P4, L18: the fact that fine textured soils can hold more water than coarse textured soils was well known before Rosenthal (2005) P5, L14: "that are differently..." P5, L21: the long-term perspective is not covered in this work, so the suggested measures may be consistent with the findings, but do not take into account climate or land use change. P6, L10: "As the distance... increases, water..." P7, L14: if I understand the sampling design correctly, there are five replicate gradients (transects perpendicular to the river), each with 6 sampling points – perhaps re-phrase? P8, L10: is the importance value calculated for each plant functional type as written, or for each species? P8, L14: RF is not present in the equations. P8, L23: the thickness of the canopy layer might not tell much about the actual biomass. Perhaps leaf area would be more representative. P8, L25: what does "them" refer to? P9, L2: suggested rephrase: "... and herb layer, which can be calculated..." Equations 5-8: to calculate D the only equation needed is Eq. 6, but in that equation, what is P? Is P related to IV defined in the previous page? Presented in this way, the equations do not seem to be related to D, which is the variable that needs to be calculated (if I understood the rationale). P9, L17: the layers used for gravimetric water content are not consistent with the layers used for other analyses P16, L22: it is not entirely clear which parameters are being predicted here – presence/absence for a given species, or the diversity indices? P18, L9: suggested rephrase: "... formed a bimodal pattern and reached local maxima at the distance..." P20, L14: "and possibly inducing..." P20, L15: as explained in the major issues above, it is not easy to infer water availability effects on the plant community from a one-time water content measurement. P20, L18:

suggested rephrase: "... also partly explained the variance of the plant community, with TP representing 8.1% of the explained variance and SOM being negatively... " P20, L22: when the groundwater table is "low", shouldn't it be "below" the degradation threshold? P20, L24: what is the relation between TP and groundwater level? P21, L4: "thus halting..." P21, L7: it is also possible that the points now at 1000 m from the river have been less disturbed, and thus harbour a community with larger biomass, diversity, or coverage.

---

## Referee Comment (RC2) · Anonymous Referee #2 · 21 Jun 2016

General Comment: Desert riparian forests are highly fragile ecosystem to climate and environmental changes. On the other hand, they serve as a haven for deteriorating desert ecosystems until their being threatened by impacts of changes. And, it is timely and relevant to have many studies on desert riparian vegetation ecology and function, as this one. The paper is well-structured and written, as well.

However, the introduction lacks a clear definition of a problem. The introduction is full of background information; like, what has been done and what is already there... Such statements cannot justify a problem of a scientific work. There has to be a strong

explanation of gaps, drawbacks... those pertinent to the subject of the work. Moreover, simple richness and classification analysis could imply "the same old story". It feels to me that more can be done with existing data beyond analysis of richness and diversity of riparian vegetation. An example is combining with current affairs like climate and environmental change, resilience, elasticity...

What is "low reaches"? This is not a professional wording; better to use simply "oasis" or "downstream", or give explanation for what "low reaches" is. It has to keep consistency, as well, in some places printed as "lower"?

The method needs more explanation how all the sampling and data collection was accomplished in one month (July 2015). Quadrants were set for collection of data on herbaceous vegetation just after the rain; what about desert-herbs those can be found before the rain?

Detail comments: Title: delete the second "soil" in the second line Page 2 Line 3: delete "of" Page 6 line 8: "this area", which area? Page 6 line 17-18: please provide professional soil-type names; "grey desert soil" is not in nomenclature of soils Page 7 line 4-6: seems part of introduction, not methods Page 7 line 9: please define what "desert riparian forests" are in your research area? Page 8 line 1-8: preferable to put in table Page 10 line 1-14: Why Monte Carlo run needed? Can't Principal Component Analysis handle that size of data? Page 10 line 17-22: please give details of TWINSPAN analysis in methods Page 10 line 27-28: what does disturbed community mean? Page 13: Figure 3 and 4 can be combined Page 14 line 2: We know for what SWC stands for, what are those attached numbers stands for? Ok, it is given in the caption, but, is also needed in the main text. Page 17: the need for Table 3 is clear; why is Table 2 (marginal and conditional effects are not main target of the study) Page 18 line 17-18: the peculiar result from the vegetation analysis is the bi-modal distribution; do the soil properties show the same pattern; so that to say "variation in soil properties...." Page 19 line 22-24: YES! this can be an inference to the bi-modal distribution (in reference to the above comment) Page 21 line 11-12: Here it says "vicinity to the main roads"?

In the methods, it is indicated sampling was done far from roads; explain why? Page 21 line 12-15: To give management options for livestock control; there is a need to have socio-economic background information, specifically to livestock, somewhere in the introduction or in methods. Page 22 line 4: what are "artificial channels"? or take the whole sentence to Introduction; also line 6, if human disturbance is a problem give a brief background in the Introduction Page 22 line 21-26: many more services can be told

---

## Author Comment (AC1) · 1 Aug 2016

We thank reviewer for the detailed comments. We have gone through all the comments and will amend the original manuscript based on the suggestions and comments. In the following pages we provide brief answers to the review comments and we will make corresponding changes after we receive the editorial decision.

Major issues

Reviewer: This work presents soil water content and biogeochemical data to explain how riparian vegetation changes as distance from the river increases. Vegetation is

characterized by species composition and diversity, and occurrence and coverage of different plant functional types. The topic is overall relevant for readers of HESS. The manuscript is relatively clear, but might benefit from proofreading by a native English speaker. Despite the interest of the topic, I have some concerns regarding the analyses conducted and the mismatch between the ecological processes causing the observed vegetation patterns, and the one-time soil sampling adopted for this study.

Authors: We thank the reviewer for the suggestions to improve the quality of this manuscript. We will carefully amend the manuscript based on the comments that you provided.

Reviewer: 1) Ecological processes vs. one-time sampling. The plant communities examined in this work are the result of decade- if not century-long successional dynamics, but they are treated as if they are the result of short term processes. I refer specifically to soil water content, used as a predictor of vegetation community despite being measured only once. How representative are these water content measurements of the long-term water availability? Other soil properties vary at slower rates and could be more meaningful predictors (texture, SOM).

Authors: We thank the reviewer for pointing out the mismatch between the ecological processes and the one-time soil sampling which is central to our study. The distribution of community is a result of long-term mutual effect between vegetation and soil. While the analysis of the temporal variance of vegetation communities may better illustrate the ecological process in a study area, it is also considered as a difficult and onerous work. We therefore propose an analysis on the temporal variation of vegetation and its relationship with environment factors using remote sensing data that can be obtain from the West Data Centre (http://westdc.westgis.ac.cn/). Although desert riparian forest community especially trees are established for decades or even longer, in our case, the growth vitality of the community and vegetation characteristics mainly formed under the influence of the ecological water conveyance which was implemented in 2000 (Zhang et al., Environmental Geology, 2009 and Zhang et al., Hydrological Processes,

2011). This program revived the whole ecosystem from severe drought and vegetation degradation after the implement of ecological water conveyance. In order to add the analysis of temporal variation, we will add the hydrological data (e.g. the amount of water conveyance, annual groundwater table) and remote sensing data (e.g. soil moisture inversion product, NDVI, LAI) to better illustrate the ecological process of community formation in the study region between 2000 and 2013. We also agree that soil moisture is much variable comparing to the soil physical and chemical properties due to the diurnal variation and annual variance. However, the dynamic monitoring data of soil moisture in the desert riparian forest showed that the diurnal variation of soil moisture was mainly restricted to the top soil layer (0-20cm), while the deeper layer was almost constant during the same day. Monitoring data also showed that soil moisture formed similar annual variation pattern due to the regulated ecological water conveyance. The soil moisture formed a unimodal pattern and peaked in July, which indicated that the soil moisture in July could reflect water condition of the community for the whole year. Thus, our sampling data of 0-200cm soil moisture is relative stable in the terms of diurnal variation and sampling time in July can represent a relatively good water condition of the site that support most vegetation communities during the year. To support this account, we will add the annual soil moisture dynamics in the supplementary material based on the dynamic monitoring data of soil moisture in desert riparian forests site.

Reviewer: How old are the trees and shrubs in this community? Are these communities shaped by the time they spent growing on a given soil (no information is provided to this regard), or by the edaphic properties of a given site (focus of the current study)?

Authors: We obtained the community age by referring to the studies on the growth characteristics of shrub and trees in the study area and consulting the local forestry government (Xiao et al., Acta Botanica Boreali-Occidentalia Sinica, 2005). The trees established on the sites are beyond 50-60 years old, while the shrubs are quite young with 80% of them developed within the last 15 years. The remaining 20% are between 15 and 30 years of age. Although trees and few shrubs initially grew on the stand in

1950s-1980s, they were in poor growing condition due to the scarce water supply from the dry stream channel (Guo et al., Environment Geology, 2008). The present growth vitality of the community and vegetation characteristics mainly formed after 2000, when the ecological water conveyance was implemented to restore the ecosystem that suffered from the severe drought at the downstream of Heihe. In addition, during the last 15 years, the large-scale factors (i.e., climate) did not change significantly (Zhang et al., Arid Land Geography, 2011). The communities in our study site are thus mainly affected by the edaphic properties rather than by time. The water condition and soil properties (soil texture, soil chemical) in our sampling are heterogeneous (Fig 5 in the manuscript), changing from shruby meadow soil to grey–brown desert soils, and finally to aeolian soil along the distance from river channel. Under a given site, the mutual effect between edaphic properties and vegetation result in the formation of certain community, and eventually result in the distribution pattern of the region. Following the question raised by reviewer, we will further add explanation regarding the formation of communities in both Methods and Discussion sections to better illustrate our points.

Reviewer: No data are reported on the variability in river discharge – how dynamic is the riparian environment? How frequent are flooding events that can re-shape the community (and soil properties)? Without this information, it is difficult to disentangle time effects from site effects.

Authors: We thank the reviewer for the question. In the downstream of Heihe, the river discharge comes from the ecological water conveyance from the middle reaches. The ecological water conveyance is an ecological restoration project conducted by the national government with the aim of restoring the ecosystems of Heihe River basin since 2000. It is implemented according to the water dispatching scheme and conducted in the April, July, August, September and November with scheduled discharge (Feng et al., Science Press Ltd, 2015). Due to the regulated water discharge, the ecological water conveyance hardly caused any flooding event. Even when a flood event happens, it only affects the sites that near the river bank (within 100m radius) (Liu et al., Beijing

Forestry University, 2011). It is unlikely to re-shape the community and soil properties of our sampling plots that mainly located beyond 100 m from the river channel. Following the reviewer's suggestion, we will add this information and data of the ecological water conveyance in the Method section to illustrate the variability of river discharge.

Reviewer: Many of the measurements used as predictors are partly correlated, making it difficult to interpret the regression results. For example, soil water content is related to texture (as noted in P4, L18). Fine textured soils can hold more water, and this effect would appear in the gravimetric water content measurements. Total nutrient (TN and TP, which I assume include organic N and P) are also correlated to SOM, since large SOM stocks are associated with large N and P stocks (as noted in P20, L30). Due to these correlations, it seems difficult to apply regression approaches that assume independence, as in this case (if I interpreted the approach correctly).

Authors: We agree with the reviewer that the factors we chose were partly correlated, such as soil water content and soil texture, TN/ TP and soil organic matter. We selected these factors that covered the aspects of soil moisture and soil properties to better illustrate the relationship between vegetation and soil in the desert riparian forest. The regression approach as mentioned by reviewer is actually a forward selection (Table 2) in the RDA (Redundancy Analysis). The RDA is an ordination rather than a regression analysis. Its main aim is sorting the principal components and finding variables that best explain the vegetation distribution. Although these factors are partly correlated, the aim of the forward selection is to identify the significant factors and their contribution rate from each principal component rather than to form the regression equation for predicting the vegetation characteristics. According to the main purpose and function of the RDA, we believe that it is reasonable to involve factors that not totally independent from each other (Lepx et al., Cambridge University Press, 2003). We will add the explanation of the RDA method in the section of Methods to clearly illustrate the analysis we used in the manuscript.

Reviewer: The conclusions are based on too short-term a study to be really useful for

planning. Either a long-term monitoring or a different study to identify possible historical reasons for the observed patterns would provide (or not!) support to a possibly large and expensive conservation project.

Authors: As suggested by the reviewer, we will add temporal analysis on the vegetation and its influencing factors during 2000-2013 to better illustrate vegetation variance after the implementation of ecological water conveyance in 2000. We will re-evaluate our conclusions after the remote sensing analysis of long-term data. By combining the temporal and spatial analysis of vegetation variance in the desert riparian forest, we would provide better suggestions to the conservation project in the study area.

Minor issues Reviewer: I am listing here only some of the small editorial issues in this MS – better to ask a native English speaker to give a thorough proofreading.

Authors: We will carefully amend the manuscript based on the editorial issues that you provided and give a thorough proofreading accordingly.

Reviewer: P2, L3: "focused" rather than "stressed"

Authors: We will replace the "stressed" with the "focused" at the P2, L3.

Reviewer: P2, L11: optimum in which sense? Is biomass higher around 1000 m, or what criteria was used to establish what the 'best' conditions are?

Authors: The "optimum" is a comprehensive demonstration of the community and environment condition based on the Shannon-Wiener diversity index, Pielou evenness index, Patrick richness index that peaked at the distance of 1000 m from the river channel. We also used other indicators. For instance, community I and II that mainly distributed within 1000 m from river formed multiple layers of vertical structure. The average community coverage also reached the highest point (88%) at this distance. Soil properties such as low bulk density and high proportion of clay contribute to the aggregation of nutrient and the transportation of soil moisture. Overall, sufficient water source, relatively good soil properties and high value of community characteristic are

the reasons we define the 1000 m away from river as the optimum range. We will add the explanation of the "best condition" in the discussion section.

Reviewer: P2, L19-20: it would be better to write if the mentioned influences are positive or negative.

Authors: We will add the positive/negative influence of soil physical and soil nutrition on the community characteristics.

Reviewer: P3, L3: vague – what ecosystem services are important in this specific context?

Authors: We will specify the ecosystem service such as sand fixation and carbon sequestration service in this study.

Reviewer: P3, L14: the term "ecological water conveyance" is not entirely clear? Is there a more commonly used term?

Authors: The "ecological water conveyance" is a restoration project with delivering the water from the middle reaches of Heihe to the low reaches of Heihe to restore the ecosystem in the low reaches which suffered from the drought stress and vegetation degradation severely. This term appeared in some relevant papers. We will explain the term in the Introduction section to make it clearer.

Reviewer: P4, L18: the fact that fine textured soils can hold more water than coarse textured soils was well known before Rosenthal (2005)

Authors: We will replace this citation with a more suitable one.

Reviewer: P5, L14: "that are differently..."

Authors: We will revise this sentence carefully according to the reviewer's suggestion

Reviewer: P5, L21: the long-term perspective is not covered in this work, so the suggested measures may be consistent with the findings, but do not take into account

climate or land use change.

Authors: Combining reviewer's suggestion on the long-term study, we will add temporal analysis on the vegetation and its influencing factors from 2000 to 2013, during which the area has experienced land use change due to the vegetation restoration and farmland expansion after implement of ecological water conveyance. This new analysis can illustrate the vegetation variance accompanied with land use change in a relative long term. The sampling data mainly illustrate the distribution pattern of desert riparian forest along the decrease gradient of water availability (i.e. the distance from the river), which may provide reference to the vegetation pattern in the drought scenario under the impact of the climate change. Based on the temporal and spatial analysis of vegetation variation, we will develop more complete suggestions on management. We will add some relevant studies in this area to support our suggestion on the management and we will rewrite this part of discussion to avoid the mismatch between our result and the discussion.

Reviewer: P6, L10: "As the distance. . .increases, water. . ."

Authors: We will revise the grammatical error of this sentence according to the reviewer's suggestion.

Reviewer: P7, L14: if I understand the sampling design correctly, there are five replicate gradients (transects perpendicular to the river), each with 6 sampling points – perhaps re-phrase?

Authors: Following the reviewer's suggestion, we will rephrase this sentence to make it clearer and easier to understand.

Reviewer: P8, L10: is the importance value calculated for each plant functional type as written, or for each species?

Authors: The importance value is calculated for each species (19 species in total). We will rephrase this sentence to make it clearer and easier to understand.

Reviewer: P8, L14: RF is not present in the equations.

Authors: We will revise this mistake and carefully check throughout the manuscript.

Reviewer: P8, L23: the thickness of the canopy layer might not tell much about the actual biomass. Perhaps leaf area would be more representative.

Authors: We agree that leaf area is better than the thickness of canopy in depicting vegetation biomass. Due to the harsh environment, however, it is more difficult to get a precise measurement of the leaf area of all species in the community because some kind of leaf turn into the assimilating branches (i.e. T. ramosissima). By contrast, the thickness of each layer is much easier to be measured and the equation of community diversity is well-reported in the literature (P27, L13: Zhu, et al, 2013).

Reviewer: P8, L25: what does "them" refer to?

Authors: It refers to the different growth type (tree layer, shrub layer, herb layer). We will rephrase this sentence to make it much clear and easier to understand.

Reviewer: P9, L2: suggested rephrase: "...and herb layer, which can be calculated..."

Authors: Following the reviewer's suggestion, we will rephrase this sentence to make it clearer and easier to understand.

Reviewer: Equations 5-8: to calculate D the only equation needed is Eq. 6, but in that equation, what is P? Is P related to IV defined in the previous page? Presented in this way, the equations do not seem to be related to D, which is the variable that needs to be calculated (if I understood the rationale).

Authors: The P refers to the important value of species (P9, L13). We apologize for using "D" in equation 4 and 6, which caused a misinterpretation of the latter. We will replace the D in P8 with a different letter to eliminate this error.

Reviewer: P9, L17: the layers used for gravimetric water content are not consistent with the layers used for other analyses

Authors: We thank the reviewer for pointing this out. Indeed, the layers used for soil moisture measurement are different from the layer used for measuring other soil properties. We divided the soil moisture into three layers in accordance with the fine root distribution of herb, shrub and tree since different layer of soil moisture showed different influence on the herb, shrub and tree in this area (The result of correlation in Table 1 showed that SWC1 mainly correlated with herb, while SWC2 and SWC3 mainly correlated with community coverage and density). The other soil properties, however, were analyzed using the mean values of each property from 0-100cm layer because the vertical variation of soil chemical properties was not significant in the data preprocessing. Thus we use different layers in analyzing soil moisture and other soil properties. We will explain this reason in the Methods section.

Reviewer: P16, L22: it is not entirely clear which parameters are being predicted here – presence/absence for a given species, or the diversity indices?

Authors: The parameters being predicted here are the community characteristics, namely the vegetation indices in Table 1. We will explain it clearly in this section to make the manuscript easier to understand.

Reviewer: P18, L9: suggested rephrase: ". . . formed a bimodal pattern and reached local maxima at the distance. . ."

Authors: Following the reviewer's suggestion, we will rephrase this sentence.

Reviewer: P20, L14: "and possibly inducing. . ."

Authors: Following the reviewer's suggestion, we will carefully revise this sentence.

Reviewer: P20, L15: as explained in the major issues above, it is not easy to infer water availability effects on the plant community from a one-time water content measurement.

Authors: Following the reviewer's suggestion, the long-term variance of water availability will be illustrated by adding the temporal analysis of soil moisture and groundwater based on the hydrological data and remote sensing data. We believe that it will clarify

the long-term water availability effects on plant community and we will add relevant discussion in the manuscript.

Reviewer: P20, L18: suggested rephrase: ". . .also partly explained the variance of the plant community, with TP representing 8.1% of the explained variance and SOM being negatively. . ."

Authors: Following the reviewer's suggestion, we will rephrase this sentence to make it clearer and easier to understand.

Reviewer: P20, L22: when the groundwater table is "low", shouldn't it be "below" the degradation threshold?

Authors: We intended to use the "low" to express the meaning of "shallow", we will replace it with "shallow" to avoid further confusion.

Reviewer: P20, L24: what is the relation between TP and groundwater level?

Authors: We did not show the relationship between TP and groundwater level directly in this study. We referred to a study reporting that the effect of TP on the vegetation was more obvious with the rapid decrease of groundwater table (P20, L25: Zhang et al., 2015b). In other words, TP exerted more influence on vegetation under the drought stress condition, which was different from our study. To avoid further confusion, we will explain it in details (e.g., from the physiological-ecological process of vegetation) in the manuscript.

Reviewer: P21, L4: "thus halting. . ."

Authors: Following the reviewer's suggestion, we will carefully revise this sentence.

Reviewer: P21, L7: it is also possible that the points now at 1000 m from the river have been less disturbed, and thus harbor a community with larger biomass, diversity, or coverage.

Authors: Thank the reviewer for pointing the possibility. In fact, the area that distance

1000m from the river is vicinity to the main road which was developed parallel to the river channel. Although the vegetation community growing nearby the road is unlikely to be disturbed by severe human as the road is separated from the surrounding by iron wire, the points at 1000 m from the river are unlikely to be less disturbed compared to other points. We will add the main road in the Figure 1 and explain the information in the Discussion section.

---

## Author Comment (AC2) · 1 Aug 2016

We thank reviewer for the detailed comments. We have gone through all the comments and will amend the original manuscript based on the suggestions and comments. In the following pages we provide brief answers to the review comments and we will make corresponding changes after we receive the editorial decision.

General Comment: Reviewer: Desert riparian forests are highly fragile ecosystem to climate and environmental changes. On the other hand, they serve as a haven for deteriorating desert ecosystems until their being threatened by impacts of changes. And,

it is timely and relevant to have many studies on desert riparian vegetation ecology and function, as this one. The paper is well-structured and written, as well. However, the introduction lacks a clear definition of a problem. The introduction is full of background information; like, what has been done and what is already there. . .Such statements cannot justify a problem of a scientific work. There has to be a strong explanation of gaps, drawbacks. . .those pertinent to the subject of the work.

Authors: We thank the reviewer's suggestion and we will carefully rewrite the Introduction section to clearly emphasize the knowledge gaps (i.e. the distribution range of desert riparian forests, the influence of soil properties on the desert riparian forests and the variation pattern of biodiversity along water availability gradients), which will help justify the problem of our scientific works.

Reviewer: Moreover, simple richness and classification analysis could imply "the same old story". It feels to me that more can be done with existing data beyond analysis of richness and diversity of riparian vegetation. An example is combining with current affairs like climate and environmental change, resilience, elasticity. . .

Authors: We thank the reviewer's constructive suggestion on further delving into the data. As suggested by the reviewer, we will try to include climate change, environmental change, resilience, and diversity indices into our revision. As community characteristics (i.e. species richness and community diversity) could improve community stability during a disturbance, these factors are usually used as indicators for vegetation resilience (Fischer et al., Philosophical Transactions B, 2016 and Samantha et al., Journal of Vegetation Science, 2016). In our sampling, the desert riparian forests were distributed along the gradient of water availability (the distance from river channel). Research on the variation of community characteristics (i.e. species richness and community diversity) along the decrease of water availability could help to discuss the vegetation resilience with the increasing drought stress and further provide reference for the ecological management under drought scenario caused by climate change. In our revision, we will recreate the Fig 3-4 and add the comparison of community characteristics among different gradients of water availability in the Results section. Furthermore, we will add discussion on the variation of vegetation resilience along the water availability gradient and provide management suggestion under changing environment by redeveloping the first and third section of the Discussion.

Reviewer: What is "low reaches"? This is not a professional wording; better to use simply "oasis" or "downstream", or give explanation for what "low reaches" is. It has to keep consistency, as well, in some places printed as "lower"?

Authors: Following the reviewer's suggestion, we will replace the "low reaches" with the word "downstream" throughout the manuscript to make the description accurate and consistent.

Reviewer: The method needs more explanation how all the sampling and data collection was accomplished in one month (July 2015). Quadrants were set for collection of data on herbaceous vegetation just after the rain; what about desert-herbs those can be found before the rain?

Authors: We thank the reviewer for the suggestion. Water availability in our study site was greatly affected by regulated water conveyance (which we will add in the Methods section) rather than the scarce precipitation of the region. Our study area belongs to hyperarid zone with mean annual precipitation below 39 mm and only 9.11 mm falls in July. There was only one rainy day on July 21 2015 when we conducted the sampling from July 10 2015 to July 30 2015. The surface soil quickly dried up before the next day due to the high evaporation (approximately 600 mm during the July). Water conveyance in the early July was therefore the only source of water for the area. Based on this condition, germination of desert herb merely benefit from the scarce precipitation, so we did not take into account the desert-herbs that can be found before the rain. As suggested by the reviewer, we will add the explanation on sampling and data collection in the Methods section.

Detail comments:

Reviewer: Title: delete the second "soil" in the second line

Authors: Following the reviewer's suggestion, we will delete this word.

Reviewer: Page 2 Line 3: delete "of"

Authors: Following the reviewer's suggestion, we will delete this word.

Reviewer: Page 6 line 8: "this area", which area?

Authors: The "this area" means the downstream of Heihe River. We will replace "this area" with "the downstream of Heihe river" to make it more clear.

Reviewer: Page 6 line 17-18: please provide professional soil-type names; "grey desert soil" is not in nomenclature of soils.

Authors: We will replace "grey desert soil" with professional soil-type names (i.e. shruby meadow soil and aeolian soil)

Reviewer: Page 7 line 4-6: seems part of introduction, not methods

Authors: We will delete these lines.

Reviewer: Page 7 line 9: please define what "desert riparian forests" are in your research area?

Authors: As suggested by the reviewer, we will add the detailed definition of "desert riparian forests" in the downstream of Heihe.

Reviewer: Page 8 line 1-8: preferable to put in table

Authors: Following the reviewer's suggestion, we will put Page 8 line 1-8 in the table annotation.

Reviewer: Page 10 line 1-14: Why Monte Carlo run needed? Can't Principal Component Analysis handle that size of data?

Authors: The Monte Carlo forward selection is a part of RDA (Redundancy Analysis).

The RDA is an ordination analysis with the aim of finding variables as the best predictors for the vegetation distribution. As the Monte Carlo forward selection can directly shows significance and contribution rate of each factor (Lepx et al., Cambridge University Press, 2003), we chose this method rather than the Principal Component Analysis (PCA). We will explain it more clearly in the Methods section.

Reviewer: Page 10 line 17-22: please give details of TWINSPAN analysis in methods

Authors: As suggested by the reviewer, we will provide detailed description of the method of TWINSPAN analysis in Methods section.

Reviewer: Page 10 line 27-28: what does disturbed community mean?

Authors: The "disturbed" is actually "distributed", we will check and revise it.

Reviewer: Page 13: Figure 3 and 4 can be combined

Authors: Following the reviewer's suggestion, we will combine these two figures into one figure.

Reviewer: Page 14 line 2: We know for what SWC stands for, what are those attached numbers stands for? Ok, it is given in the caption, but, is also needed in the main text.

Authors: As suggested by the reviewer, we will add the explanation of SWC when it appears in the main text.

Reviewer: Page 17: the need for Table 3 is clear; why is Table 2 (marginal and conditional effects are not main target of the study)

Authors: Table 2 allows the selection of the key influencing factors from the marginal and conditional effects. Marginal effects reflected the effects of the environmental variable on the community characteristics, while conditional effects reflected the effects of the environmental variables on the community characteristics after the anterior variable was eliminated by the forward selection method. The forward selection in the Table 2 allowed key variables to be determined through the strength of their effects and significance. Based on the key influencing factors selected in the Table 2, we further analyze the variation of community characteristics explained by different groups of key environmental factors, which is the result showed in Table 3. We will revise the title of Table 2 and explain the purpose of Table 2 in the result part to express the meaning clearly.

Reviewer: Page 18 line 17-18: the peculiar result from the vegetation analysis is the bi-modal distribution; do the soil properties show the same pattern; so that to say "variation in soil properties. . . ." Page 19 line 22-24: YES! this can be an inference to the bi-modal distribution (in reference to the above comment)

Authors: We thank the reviewer's suggestion on improving our Discussion on how "variation in soil properties" may affect vegetation community. We will further develop this topic by referring to the results in Page 19 line 22-24.

Reviewer: Page 21 line 11-12: Here it says "vicinity to the main roads"? In the methods, it is indicated sampling was done far from roads; explain why?

Authors: In the Methods section, the description of "we chose sites that were far from farmlands, roads, irrigated channels and reservoirs" means that the general principle we choose plots is distant from the roads and paths as far as possible to minimize the human disturbance (i.e. grazing and firewood cutting) on the vegetation communities. However, in the study area, there is a main road extending across the oasis and almost parallel to the river channel. As the distance of each sampling plots from the river channel is fixed, it is impossible to avoid sampling near the main road which extents parallel to the river channel. Currently, the vegetation community growing nearby the road is unlikely to be disturbed by human as the road is separated from the surrounding by iron wire. So sampling near the main road does not go against with our general principle to minimize the human disturbance (i.e. grazing and firewood cutting) on the vegetation communities. In addition, there are only 4 points located within 300m from the main road, thus the data obtained from those points did not affect the overall analysis result. But the vegetation communities growing nearby the main road may be disturbed by

human activity in the future due to increasing population living and traveling there. The living range (1000m from river channel) mentioned in Page21 line11-12 is located near to the main road, which means this gradient (1000m from river channel) is likely to be disturbed by human activity in the future due to its easy accessibility. So we described the area that distant 1000m from river channel as "vicinity to the main roads" and listed the possible human influence on this gradient in the following sentences. We will explain it clearly in the Method part and add the main road in the Figure 1 to make it easier to be understood.

Reviewer: Page 21 line 12-15: To give management options for livestock control; there is a need to have socio-economic background information, specifically to livestock, somewhere in the introduction or in methods.

Authors: Following the reviewer's suggestion, we will add the socio-economic background information in the introduction to make this discussion part more complete.

Reviewer: Page 22 line 4: what are "artificial channels"? or take the whole sentence to Introduction; also line 6, if human disturbance is a problem give a brief background in the Introduction

Authors: The "artificial channels" means that the river channel built perpendicular to the river with the aim of delivering the water for irrigation. The channel was built by the concrete which generating little benefit to these vegetation communities, comparing the natural channels that have the seepage property. As we chose points that distant from the roads, farmlands and irrigated channels as far as possible to minimize the human disturbance, the human disturbance mentioned in line 6 is not the main factor that shaped the vegetation community. It is the possible disturbance on the vegetation community that needs to be considered in the future management. As suggested by the reviewer, we will put the sentence of "artificial channel" in the Introduction and add the information of possible human disturbance in the Introduction part.

Reviewer: Page 22 line 21-26: many more services can be told

Authors: Following the reviewer's suggestion, we will add discussion on more ecosystem service, such as sand fixation, water conservation and carbon sequestration in this discussion part to fully illustrate the importance of conserving ecosystem.
* * *

---

## Author Response (AR1)

**Responses to the Editor and Reviewers:**

We thank the editor and reviewers very much for the time they spent evaluating our manuscript and providing constructive comments. Their detailed comments inspired us to improve the quality of this manuscript. We have gone through all the comments and amended the original manuscript based on the suggestions and comments. In the following pages we provide point-by-point responses to the editor's and reviewers' comments. Please refer to the attached manuscript with track-changes mode for further details.

**Responses to the Editor:**

Editor: The authors may have an interesting story to tell, but both reviewers were in agreement that major revisions are needed if this is to be acceptable for publication. All of the major the major comments by reviewer #1 are of particular importance to address, even though all of the points by both reviewers should be addressed in a revision

Authors: We thank the editor for the pointing out the importance of the major comments of reviewer #1, we have carefully addressed all the major comments by reviewer #1 in the response to reviewer #1. We have carefully addressed all the comments by both reviewers in this response and amended the original manuscript based on the suggestions and comments. In addition, we added Yaping Wang as the co-author of this manuscript for her contribution in adding temporal analysis part in this revised manuscript.

**Responses to the Reviwer#1:**

**Major** issues**

Reviewer: This work presents soil water content and biogeochemical data to explain how riparian vegetation changes as distance from the river increases. Vegetation is characterized by species composition and diversity, and occurrence and coverage of different plant functional types. The topic is overall relevant for readers of HESS. The manuscript is relatively clear, but might benefit from proofreading by a native English speaker. Despite the interest of the topic, I have some concerns regarding the analyses conducted and the mismatch between the ecological processes causing the observed vegetation patterns, and the one-time soil sampling adopted for this study.

Authors: We thank the reviewer for the constructive suggestions to improve the quality of this manuscript. We have carefully amended the manuscript by adding the temporal analysis to address the questions. We illustrated the temporal variation of vegetation communities by adding the temporal variation of NDVI during 2000-2014 based on the Landsat TM/ETM

(30m) image (please see Page 18 Line 2 - 8 in result part and Page 26 Line 20 - Page 29 Line 6 in the Discussion section). We further added the temporal variation factors (e.g., groundwater, soil moisture, runoff) in the environment factors to illustrate how spatial heterogeneity and temporal variation factors drive the variation of desert riparian forest (please see "3.4 Pearson correlation between community characteristics and environmental factors" at Page 20 Line 15 - Page 21 Line 19 and "3.5 Key environmental factors that influenced community characteristics" at Page 23 Line 2 - 16 in the Result section; Page 29 Line17 - Page30 Line 7 in the Discussion section). Then, based on the analysis of spatial distribution and temporal variation of desert riparian forest, we proposed suggestions for the restoration under the changing environment (please see Page 31 Line14 - Page 32 Line 4).

**Reviewer:**

**1) Ecological processes vs. one-time sampling.**

The plant communities examined in this work are the result of decade- if not century-long successional dynamics, but they are treated as if they are the result of short term processes. I refer specifically to soil water content, used as a predictor of vegetation community despite being measured only once. How representative are these water content measurements of the long-term water availability? Other soil properties vary at slower rates and could be more meaningful predictors (texture, SOM).

Authors: We thank the reviewer for pointing out the discrepancy between the successional dynamics of ecological processes and one-time sampling. Although desert riparian forest communities especially trees have been established for decades or even longer, in our case, the recovery of the community and vegetation characteristics were mainly formed under the influence of the ecological water conveyance implemented in 2000 (Zhang et al., Hydrological Processes, 2011). Because there has been no long-term field data to monitor the vegetation communities since the implementation of ecological water conveyance, we used high-resolution remote sensing image to analyze the long-term temporal vegetation variation in restored area. We added the analysis on the temporal variation of NDVI at our sampling sites based on the Landsat TM/ETM (30m) image since the implementation of ecological water conveyance (from 2000 to 2014) (please see Page 18 Line 2 - 8 in result part and Page 26 Line 20 - Page 29 Line 6 in the Discussion section).

In addition, we added the annual average and change rate of 2 cm soil moisture, 100 cm soil moisture, groundwater and runoff from 2000-2013 based on the retrieved remote sensing data as temporal variation factors to fully address the impact of water availability on the desert riparian forest (please see Page 20 Line 4 - Line 10 in the Result section). We further examined the impact of spatial heterogeneity and temporal variation on the variation of desert

riparian forest in the restoration and disentangled the contribution of each factor on vegetation variance (please see "3.4 Pearson correlation between community characteristics and environmental factors" at Page 20 Line 15 - Page 21 Line 19 and "3.5 Key environmental factors that influenced community characteristics" at Page 23 Line 2 – Line 16 in the Result section; Page 27 Line 10 – Page 30 Line 7 in the Discussion section). The temporal variation of soil moisture derived from retrieved remote sensing data since 2008 showed that the soil moisture was relatively stable (please see Page 20, Fig. 7a, b in the Result section) particularly at the deeper soil layers (below 20 cm), which could represent the water condition at the sampling site. Monitoring data also showed that soil moisture in July could reflect the best water condition of the community for the whole year (Fig. S1 in the supplementary material). Thus, our sampling data in July using 0-200cm soil moisture could represent a relatively good water condition of the site that supported most vegetation communities after 15 years' restoration.

Reviewer: How old are the trees and shrubs in this community? Are these communities shaped by the time they spent growing on a given soil (no information is provided to this regard), or by the edaphic properties of a given site (focus of the current study)?

Authors: We obtained the community age by referring to the studies on the growth characteristics of shrub and trees in the study area and consulting the local forestry government (Xiao et al., Acta Botanica Boreali-Occidentalia Sinica, 2005). The trees established on the sites are beyond 50-60 years old, while the shrubs are quite young with 80% of them developed within the last 15 years. The remaining 20% are between 15 and 30 years of age. Although trees and few shrubs initially grew on the stand in 1950s-1980s, they were in poor growing condition due to the scarce water supply from the dry stream channel (Guo et al., Environment Geology, 2009). The present community condition and characteristics were mainly formed after 2000. We analyzed the change of community composition in our manuscript (please see Page16 Line 1 – Line 9 in result part and Fig. 3 in Page16). About 60% of desert riparian forests remained unchanged with regard to community composition and the remaining mainly shifted from the sparse (e.g. sparse forest land, bareland land) to denser vegetation community (e.g. shrubland and grassland).

As for the driving factors, we analyzed both heterogeneity of soil property and temporal variation of water availability in the revised manuscript. We found that the heterogeneity of soil properties was the main driving force of the vegetation variation, accounting for 98.4% of total explanation, while temporal variation factors explained 35.9% of total explanation and these two groups of factors together accounted for 34.3% of the total explanation (please see

"3.5 Key environmental factors that influenced community characteristics" at Page 23 Line 2 – Line 16 in the Result section; "4.2 Factors influencing the distribution pattern and temporal variation of desert riparian forest" at Page 27 Line 10 – Page 30 Line 7 in the Discussion section). With regards to the fact that large-scale factors (i.e., climate) did not change significantly during the last 15 years (Zhang et al., Arid Land Geography, 2011), we concluded that the communities in our study site were mainly affected by the edaphic properties rather than by time. The water condition and soil properties (soil texture, soil chemical) in our sampling were heterogeneous (please see Page 19 Fig 6 and Page18 Line 15 -Page 19 Line 6 in the Result section), changing from shruby meadow soil to grey–brown desert soils, and finally to aeolian soil along the distance from river channel. Under a given site, the mutual effect between edaphic properties and vegetation resulted in the formation of certain community, and eventually the distribution pattern of the region.

Reviewer: No data are reported on the variability in river discharge – how dynamic is the riparian environment? How frequent are flooding events that can re-shape the community (and soil properties)? Without this information, it is difficult to disentangle time effects from site effects.

Authors: We thank the reviewer for the question. In the downstream Heihe River Basin, the ecological water conveyance delivers water downstream as part of an ecological restoration project conducted by the national government with the aim of restoring the ecosystems of the river since 2000. It is implemented according to water dispatching scheme scheduled in the April, July, August, September and November (Feng et al., Science Press Ltd, 2015). Due to the regulated water discharge, the ecological water conveyance hardly caused any flooding event. Even if a flood happened, it only affected the sites that near the river bank (within 100m radius) (Liu et al., Journal of Glaciology and Geocryology, 2008). It is unlikely to reshape the community and soil properties of our sampling plots mainly located beyond 100 m from the river channel. Following the reviewer's suggestion, we added this information of the ecological water conveyance in the Data and methods section (please see Page 9 Line 19-21). In addition, we added the runoff data which indicated the water conveyance in each year and analyzed the relationship between runoff and NDVI (please see Page 21 Line 13 – Line 19 in Result section).

Reviewer: Many of the measurements used as predictors are partly correlated, making it difficult to interpret the regression results. For example, soil water content is related to texture (as noted in P4, L18). Fine textured soils can hold more water, and this effect would appear in the gravimetric water content measurements. Total nutrient (TN and TP, which I assume include organic N and P) are also correlated to SOM, since large SOM stocks are associated

with large N and P stocks (as noted in P20, L30). Due to these correlations, it seems difficult to apply regression approaches that assume independence, as in this case (if I interpreted the approach correctly).

Authors: We agree with the reviewer that the factors we chose were partly correlated, such as soil water content and soil texture, TN/ TP and soil organic matter. We selected these factors that covered the aspects of soil moisture and soil properties to better illustrate the relationship between vegetation and soil in the desert riparian forest. Although we did use regression approach in the manuscript, it mainly used to examine variation of community characteristics along the river channel and the result was presented in Fig. 4 (please see Page 17 in the manuscript). The regression approach as mentioned by reviewer is actually a forward selection (please see Page 24 Table 2 in the manuscript) in the RDA (Redundancy Analysis). The RDA is an ordination rather than a regression analysis. Its main aim is sorting the principal components and finding variables that best explain the vegetation distribution. Although these factors are partly correlated, the aim of the forward selection is to identify the significant factors and their contribution rate from each principal component rather than to form the regression equation for predicting the vegetation characteristics. According to the main purpose and function of the RDA, we believe that it is reasonable to involve factors that not totally independent from each other (Lepš et al., Cambridge University Press, 2003). We add the explanation of the RDA in the Data and methods section to clearly illustrate the analysis we used in the manuscript. Please see Page 13 Line 21 - Line 24: "To further separating the key influencing factors of the 18 environment variables, marginal and conditional effects of various variables were calculated through the Monte Carlo forward selection in RDA (Redundancy Analysis), which directly showed the significance and contribution rate of each factor".

Reviewer: The conclusions are based on too short-term a study to be really useful for planning. Either a long-term monitoring or a different study to identify possible historical reasons for the observed patterns would provide (or not!) support to a possibly large and expensive conservation project.

Authors: We thank reviewer for this suggestion. We added the temporal variation of desert riparian forest to better illustrate how vegetation changed after the implementation of ecological water conveyance in 2000 (please see Page 18 Line 2 – Line 8 in Result section and Page 26 Line 20 - Page 27 Line 6 in Discussion section), including the combination soil spatial heterogeneity and water availability during 15 years of restoration (please see "3.3 The spatial and temporal variation of water availability and soil properties" at Page 18 Line 15 - Page 20 Line 10 and in Result section "4.2 Factors influencing the distribution pattern and

temporal variation of desert riparian forest" at Page 27 Line 10 - Page 30 Line 7 in Discussion section). We found that the despite the general increase trend of NDVI, the area within 500 m from river channel underwent degradation in recent years due to the intensive human disturbance, which called for further protection near the river bank. In addition, to address the influence of ecological conveyance, we analyze the relationship between runoff and NDVI. Apart from one-year lag found in the impact of runoff which responsible for the initial decrease of NDVI from 2000-2002 (please see Page 21 Line 13 - Line 19 in Result section), we found that soil heterogeneity accounted for most of explanation (98.4%) for vegetation recovery after long term restoration. Thus multiple conservation measures on protecting the soil structure (e.g., build artificial soil cover and livestock grazing exclusion) were recommended for this region to reduce the adverse effects of grazing on soil properties. Moreover, we discerned the community resilience in each gradient based on the distribution pattern of diversity and we proposed suggestion on the restoration under the changing environment (please see Page 30 Line 23 - Page 31 Line 9). Through the analysis of distribution pattern and temporal variation of desert riparian forest during restoration as well as discerned the key influencing factors driving the variation, our study could provide some meaningful supports to the future restoration.

**Minor issues**

Reviewer: I am listing here only some of the small editorial issues in this MS – better to ask a native English speaker to give a thorough proofreading.

Authors: We have carefully amended the manuscript based on the editorial issues that you provided and gave a thorough proofreading accordingly.

Reviewer: P2, L3: "focused" rather than "stressed"

Authors: We rewrote the abstract (please see Page 2 Line 3 – Line 6), deleted the sentence and changed it into: "Since they are also sensitive to disturbance examining the distribution pattern, temporal variation of desert riparian forest and their influencing factors are important to determine the limiting factors of vegetation recovery after long-term restoration".

Reviewer: P2, L11: optimum in which sense? Is biomass higher around 1000 m, or what criteria was used to establish what the 'best' conditions are?

Authors: We rewrote the manuscript and based the relationship between diversity indices and community resilience, the optimum range of desert riparian forest was replaced by the discussion on the community resilience in each gradient (please see Page 30 Line 23 - Page 31 Line 9).

**Reviewer: P2, L19-20: it would be better to write if the mentioned influences are positive or negative.**

Authors: We rewrote the abstract and replaced the sentence with the impact of spatial heterogeneity factors on the vegetation and illustrate the positive influence on the vegetation. Please see Page 2 Line 25 – Line 28: "Spatial heterogeneity factors, accounting for 98.4% of the total explanation, positively influenced the community diversity, structure, average NDVI and change rate of NDVI. Temporal variation factors accounting for 35.9% explanation and positively influenced the community daverage NDVI".

**Reviewer: P3, L3: vague – what ecosystem services are important in this specific context?**

Authors: We specified the ecosystem service such as sand fixation and carbon sequestration service in this study. Please see Page 3 Line 11 – Line 13: "Riparian zone is the linkage between terrestrial and aquatic ecosystem, which plays an important role in ecological processes and provides a variety ecosystem services, such as sand stabilization and carbon sequestration".

Reviewer: P3, L14: the term "ecological water conveyance" is not entirely clear? Is there a more commonly used term?

Authors: The "ecological water conveyance" is a restoration project with delivering the water from the middle reaches of Heihe to the low reaches of Heihe to restore the ecosystem in the low reaches which suffered from the drought stress and vegetation degradation severely. This term appeared in some relevant papers. We explained the term in the Introduction section. Please see Page 3 Line 23 – Line 26: "Since 2000, ecological water conveyance project (EWCP), a restoration project aimed to deliver water downstream has been implemented to restore the ecosystems of the Heihe River Basin".

Reviewer: P4, L18: the fact that fine textured soils can hold more water than coarse textured soils was well known before Rosenthal (2005).

Authors: We rewrote the Introduction and the citation was deleted in the revised manuscript.

Reviewer: P5, L14: "that are differently..."

Authors: We revised this sentence carefully according to the reviewer's suggestion. Please see Page 6 Line 29 – Line 30: "We investigated variability in desert riparian forests sites that are differently located along the perpendicular direction from the river channel".

Reviewer: P5, L21: the long-term perspective is not covered in this work, so the suggested measures may be consistent with the findings, but do not take into account climate or land use change.

Authors: Using the long-term data as suggested by the reviewer, we added the temporal variation of desert riparian forest to better illustrate how vegetation changed after the implementation of ecological water conveyance in 2000 (please see Page 18 Line 2 – Line 8 in result part and Page 26 Line 20 - Page 27 Line 6 in the Discussion section). This temporal analysis illustrated the vegetation recovery accompanied with change in vegetation composition change during the restoration (please see Page16 Line 1-9 in Result section). The sampling data mainly illustrated the distribution pattern of desert riparian forest along the decreasing gradient of water availability (i.e. the distance from the river), which may provide reference to vegetation pattern during drought with climate change scenario. Based on the temporal and spatial analysis of vegetation variation, we developed a more comprehensive suggestions on management (please see Page 31 Line 14 - Page 32 Line 4 in the Discussion section).

**Reviewer: P6, L10: "As the distance...increases, water..."**

Authors: We revised the grammatical error of this sentence according to the reviewer's suggestion. Please see Page 8 Line 10: "As the distance from river channel increases, water availability declines and the vegetation shifts from desert riparian forests to desert scrub. The desert riparian forests are the main components of Ejina oasis".

Reviewer: P7, L14: if I understand the sampling design correctly, there are five replicate gradients (transects perpendicular to the river), each with 6 sampling points – perhaps rephrase?

Authors: Consistent with reviewers' understanding, the sampling was conducted on five replicate gradients that perpendicular to the river and each with 7 sampling points that is 100 m, 500 m, 1000 m, 1500 m, 2000 m, 2500 m, and 3000 m from river channel, respectively. Following the reviewer's suggestion, we rephrased this sentence to make it clearer. Please see Page 9 Line 20 - Line 26: "Therefore, vegetation and soil samplings were conducted

perpendicular to the river channel and there were several locations based on the distance from the river channel: 100 m, 500 m, 1000 m, 1500 m, 2000 m, 2500 m, and 3000 m, respectively, generating a total of 35 sampling sites."

Reviewer: P8, L10: is the importance value calculated for each plant functional type as written, or for each species?

Authors: The importance value is calculated for each species (19 species in total). We rephrased this sentence. Please see Page11 Line 19 – Line 20: "The P (importance value) of each tree, shrub and herb in each plant site was calculated for each species using the following formulas".

**Reviewer: P8, L14: RF is not present in the equations.**

Authors: We revised this mistake and carefully checked throughout the manuscript, Please see Page 11 Line 23: "where RDen is the relative density, RF is the relative frequency, RDom is the relative dominance, RH is the relative coverage and RC is the relative coverage".

Reviewer: P8, L23: the thickness of the canopy layer might not tell much about the actual biomass. Perhaps leaf area would be more representative.

Authors: We agree that leaf area is better than the thickness of canopy in depicting vegetation biomass. Due to the harsh environment, however, it is more difficult to get a precise measurement of the leaf area of all species in the community because some leaf turn into the assimilating branches (i.e. *T. ramosissima*). By contrast, the thickness of each layer is much easier to be measured and the equation of community diversity is commonly used in the literature (Zhu, et al, Ecohydrology, 2013).

**Reviewer: P8, L25: what does "them" refer to?**

Authors: It refers to the different growth type (tree layer, shrub layer, herb layer). We rephrased this sentence in the manuscript. Please see Page12 Line 5 - Line 7: "Among different growth type, the thickness of tree leaf layer is calculated at 33.3% the height of the tree layer, the shrub layer is at 50% and the herb layer is at 100%".

**Reviewer: P9, L2: suggested rephrase: "...and herb layer, which can be calculated..."**

Authors: Following the reviewer's suggestion, we rephrased this sentence in the manuscript.

Please see Page12 Line 11 - Line 12: "*A* is the diversity index of the tree layer, shrub layer and herb layer, which can be calculated using the formulae listed below."

Reviewer: Equations 5-8: to calculate D the only equation needed is Eq. 6, but in that equation, what is P? Is P related to IV defined in the previous page? Presented in this way, the equations do not seem to be related to D, which is the variable that needs to be calculated (if I understood the rationale).

Authors: The P refers to the important value of species (P13, L5). We apologize for using "D" in equation 4 and 6, which caused a misinterpretation of the latter. We replaced the "D" with letter "A" to eliminate this error. Please see Page 12 Line 9:  ${}^{A=}\sum W_i A_i$ . We also changed the IV to P to make it consistent in the manuscript. Please see Page 11 Line 21 – Line 22: "The P (importance value) of each tree, shrub and herb in each plant site was calculated for each species using the following formulas: P Tree = (*RDen* + *RDom* + *RH*)/3 (1) P Shrub or Grass= (*RDen* + *RDom* + *RC*)/3 (2)".

Reviewer: P9, L17: the layers used for gravimetric water content are not consistent with the layers used for other analyses.

Authors: We thank the reviewer for pointing this out. Indeed, the layers used for soil moisture measurement are different from the layer used for measuring other soil properties. We divided the soil moisture into three layers in accordance with the fine root distribution of herb, shrub and tree since different layer of soil moisture showed different influence on the herb, shrub and tree in this area (The result of correlation in Table 1 showed that SWC1 mainly correlated with herb, while SWC2 and SWC3 mainly correlated with community coverage and density). The other soil properties, however, were analyzed using the mean values of each property from 0-100cm layer because the vertical variation of soil chemical properties was not significant in the data preprocessing. Thus we use different layers in analyzing soil moisture and other soil properties. We explained this reason in the Data and methods section. Please see Page 12 Line 25 - Page 13 Line 1: "To depict the vertical structure of soil moisture, soil water content was divided into three layers: 0-30 cm soil moisture (SWC1), 30-100 cm soil moisture (SWC2), and 100-200 cm soil moisture (SWC3) in accordance to the fine roots distribution of herbs, trees and shrubs in this area. We averaged the soil moisture at each corresponding finer increment to obtain the value of SWC1, SWC2 and SWC3. Soil chemical properties, however, were analyzed using the mean values of 0 -100cm due to the minor vertical variation".

Reviewer: P16, L22: it is not entirely clear which parameters are being predicted here -

**presence/absence for a given species, or the diversity indices?**

Authors: The parameters being predicted here are the community characteristics, namely the vegetation indices in Table 1. We explained it in the Result section: 3.5 Key environmental factors that influenced community characteristics. Please see Page 23 Line 2 - Line 4: "To further examine the key environmental factors that controlled the variation of vegetation indices (e.g. community diversity, structure, NDVI), redundant variables were eliminated by a forward selection method".

Reviewer: P18, L9: suggested rephrase: "... formed a bimodal pattern and reached local maxima at the distance..."

Authors: Following the reviewer's suggestion, we rephrased this sentence. Please see Page 25 Line 9: "The community coverage reached local maxima at the distance of 1000 m and 3000 m where community consisted of diverse shrub and herb layers."

**Reviewer: P20, L14: "and possibly inducing..."**

Authors: Following the reviewer's suggestion, we revised this sentence. Please see Page 28 Line 20 – Line 24: "While, soil with high bulk density often consisted of high silt and sand, but low percentage of clay which resulted in low water holding capacity in the surface soil and possibly inducing the drought stress to the vegetation community".

Reviewer: P20, L15: as explained in the major issues above, it is not easy to infer water availability effects on the plant community from a one-time water content measurement.

Authors: Following the reviewer's suggestion, the long-term variance of water availability was illustrated by adding the temporal analysis of soil moisture and groundwater based on the hydrological data (Fig. S1 in the supplementary material) and retrieved remote sensing data (Fig. 7, Page 20 in the manuscript). We analyzed the impact of temporal variation water availability and the heterogeneity of water availability on plant community in the manuscript (please see "4.2 Factors influencing the distribution pattern and temporal variation of desert riparian forest" at Page 27 Line 17 - Page 28 Line 8, Page 29 Line 17 - Page 30 Line 7 in the Discussion section).

Reviewer: P20, L18: suggested rephrase: "...also partly explained the variance of the plant community, with TP representing 8.1% of the explained variance and SOM being negatively..."

Authors: Because we added the temporal variation factors in to environment factors, we reconducting data analysis and the TP was no longer the key influencing factor, thus we deleted this sentence.

**Reviewer: P20, L22: when the groundwater table is "low", shouldn't it be "below" the degradation threshold?**

Authors: We intended to use the "low" to express the meaning of "shallow", but because of the reorganization of the Discussion section, we deleted the sentence.

**Reviewer: P20, L24: what is the relation between TP and groundwater level?**

Authors: In the original manuscript, we found that TP was one of the key factors influencing vegetation characteristics, accounting for 8.1% of variance. However, after we added the temporal variation factors in to environment factors and we re-conducted data analysis, TP only explained 2% vegetation variance and no longer became the key influencing factor (please see Table 2 in Page 24). We therefore deleted this sentence in the manuscript.

Reviewer: P21, L4: "thus halting..."

Authors: Following the reviewer's suggestion, we carefully revised this sentence. Please see Page 29 Line 15: "At the same time, the dominant species (i.e., *P. euphratica* and *T. ramosissima*), despite producing high amount of litter, they also had high competition for resources, thus halting the diversity and growth of other species".

Reviewer: P21, L7: it is also possible that the points now at 1000 m from the river have been less disturbed, and thus harbor a community with larger biomass, diversity, or coverage.

Authors: Thank the reviewer for pointing out the possibility. In fact, the 1000 m location along the gradient is the area which attract most tourists and herbivores, and consequently is more disturbed compared to other locations. Thus the larger biomass, diversity, or coverage did not result from fewer disturbances in the area.

**Responses to the Reviwer#2:**

**General Comments:**

Reviewer: Desert riparian forests are highly fragile ecosystem to climate and environmental changes. On the other hand, they serve as a haven for deteriorating desert ecosystems until their being threatened by impacts of changes. And, it is timely and relevant to have many studies on desert riparian vegetation ecology and function, as this one. The paper is well-structured and written, as well. However, the introduction lacks a clear definition of a problem. The introduction is full of background information; like, what has been done and what is already there...Such statements cannot justify a problem of a scientific work. There has to be a strong explanation of gaps, drawbacks...those pertinent to the subject of the work.

Authors: We thank the reviewer's suggestion and we carefully rewrote the Introduction section. Combing the first reviewer's suggestion on the temporal variation of the desert riparian forest, we added the temporal analysis and revised the main scientific questions in our study. In the revised manuscript, we emphasize the knowledge gaps on: 1) the influence of soil properties on the desert riparian forests rather than the groundwater focuses as in previous studies (please see Page 4 Line 13 - Line 26), 2) comprehensive analysis on the spatial and temporal variation of vegetation characteristic during the restoration process (please see Page 5 Line 17- Line 24), and 3) disentangling the impact of spatial heterogeneity factors and temporal variation factors on the vegetation communities (please see Page 5 Line 25 - Line 29). These knowledge gaps were addressed in the Introduction to justify the need of our work (please see Page 7 Line 6 - Line 13).

Reviewer: Moreover, simple richness and classification analysis could imply "the same old story". It feels to me that more can be done with existing data beyond analysis of richness and diversity of riparian vegetation. An example is combining with current affairs like climate and environmental change, resilience, elasticity...

Authors: We thank the reviewer's constructive suggestion on further delving into the data. As suggested by the reviewer, we included climate change, environmental change, resilience, and diversity indices into our revision. We rewrote the Discussion section and added the discussion on the community resilience and possible management in the "4.3 Community resilience of desert riparian forests and implications for ecological protection" (please see Page 30). Based on the relationship between community characteristics (i.e., species richness and community diversity) and community resilience, we discussed the community resilience

in each sampling gradient. We defined the distance of 1000 m and 3000 m from river channel as the critical region with high resilience against disturbance (please see Page 30 Line 23 - Page 31 Line 9). We addressed the potential threats to these locations under climate change (e.g., drought stress) and intensive human disturbance (e.g., grazing and tourism pressure) and we further proposed the possible management to effectively restore the ecosystem in the future (please see Page 31 Line 14 - Page 32 Line 4).

Reviewer: What is "low reaches"? This is not a professional wording; better to use simply "oasis" or "downstream", or give explanation for what "low reaches" is. It has to keep consistency, as well, in some places printed as "lower"?

Authors: Following the reviewer's suggestion, we replaced the "low reaches" with the word "downstream" throughout the manuscript to make the description accurate and consistent.

Reviewer: The method needs more explanation how all the sampling and data collection was accomplished in one month (July 2015). Quadrants were set for collection of data on herbaceous vegetation just after the rain; what about desert-herbs those can be found before the rain?

Authors: We thank the reviewer for the suggestion. Water availability in our study site was greatly affected by the regulated water conveyance rather than the scarce precipitation of the region. Our study area belongs to hyperarid zone with mean annual precipitation below 39 mm and only 9.11 mm falls in July. There was only one rainy day (July 21 2015) when we conducted the sampling from July 10 2015 to July 30 2015. The surface soil quickly dried up before the next day due to the high evaporation (approximately 600 mm during the July). Water conveyance in the early July was therefore the only source of water for the area. Based on this condition, germination of desert herb barely benefited from the scarce precipitation, so we did not take into account the desert-herbs that could be found before the rain. As suggested by the reviewer, we added the explanation on sampling and data collection in the Data and methods section (please see Page 9 Line 14 - Page 10 Line 3).

**Detail comments:**

**Reviewer: Title: delete the second "soil" in the second line**

Authors: Because of adding the temporal analysis of vegetation variance, we changed the title of the manuscript to "The distribution pattern and temporal variation of desert riparian forests and its influencing factors in the downstream Heihe River Basin, China", please see Page 1.

**Reviewer: Page 2 Line 3: delete "of"**

Authors: Following the reviewer's suggestion, we deleted this word.

**Reviewer: Page 6 line 8: "this area", which area?**

Authors: "this area" means the downstream Heihe River. We rewrote this sentence and delete "this area" and the revised sentence is "Due to sparse precipitation and hyperarid environment, Heihe River is the main source of recharge for the groundwater system in Ejina Oasis" (please see Page 8 Line 7 - Line 9).

**Reviewer: Page 6 line 17-18: please provide professional soil-type names; "grey desert soil" is not in nomenclature of soils.**

Authors: We replaced "grey desert soil" with professional soil-type names (i.e. shruby meadow soil and aeolian soil). We rewrote the sentence. Please see Page 8 Line 17 – Line 18: "The main soil types in the area are shruby meadow soil, aeolian soil and grey-brown desert soils. Saline-alkaline soils and swamp soils also exist in the lake basins and lowlands".

**Reviewer: Page 7 line 4-6: seems part of introduction, not methods**

Authors: Following the reviewer's suggestion, we deleted these lines.

**Reviewer: Page 7 line 9: please define what "desert riparian forests" are in your research area?**

Authors: As suggested by the reviewer, we added the detailed definition of "desert riparian forests" in the downstream Heihe River Basin. Please see Page 9 Line 2-4: "In the downstream Heihe River Basin, the desert riparian forest makes up the main body of the desert oasis, mainly comprised of tree, shrub and grass communities. The forests are distributed along the Heihe River from 0 m to 2000 m from river channel".

**Reviewer: Page 8 line 1-8: preferable to put in table**

Authors: Thanks the reviewer's suggestion, we put these lines in the annotation of Table 1 in the manuscript. Please see Page 22 Line 5 - Line 8: "0-20cm soil particle composition were analyzed in the laboratory for the silt (<0.02mm), clay (0.02-0.05 mm), sand (0.05-2 mm), and gravel (>2mm) contents by using Mastersizer 2000. Soil chemical properties at 0-20, 20-

40, 40-60, 60-80 and 80-100 cm and the average value of 0-100cm were used in the analysis".

**Reviewer: Page 10 line 1-14: Why Monte Carlo run needed? Can't Principal Component Analysis handle that size of data?**

Authors: The Monte Carlo forward selection is a part of RDA (Redundancy Analysis). The RDA is an ordination analysis with the aim of finding variables as the best predictors for the vegetation distribution. As the Monte Carlo forward selection can directly shows significance and contribution rate of each factor (Lepš et al., Cambridge University Press, 2003), we chose this method rather than the Principal Component Analysis (PCA). We added these explanations in the Data and methods section. Please see Page 13 Line 21 – Line 24: "To further separating the key influencing factors of the 18 environment variables, marginal and conditional effects of various variables were calculated through the Monte Carlo forward selection in RDA (Redundancy Analysis), which directly showed the significance and contribution rate of each factor".

**Reviewer: Page 10 line 17-22: please give details of TWINSPAN analysis in methods**

Authors: As suggested by the reviewer, we provided detailed description of the method of TWINSPAN analysis in Data and methods section. Please see Page 13 Line 15 - Line 20: "To depict the variation of desert riparian forests composition, we used Two-way Indicator Species Analysis (TWINSPAN, in WinTWINSPAN, version 2.3), a method of community hierarchical classification based on the importance value of each species (Hill, 1979), to classify the possible desert riparian forests community types. The importance value data for all plant species, obtained from the vegetation survey were used in this analysis and the cutoff levels of importance value for each class were set as: 0, 0.1, 0.2, 0.4, 0.6 and 0.9".

**Reviewer: Page 10 line 27-28: what does disturbed community mean?**

Authors: The "disturbed" is actually "distributed", we thank the reviewer for pointing out this oversight. Please see Page 14 Line 18: "This community mainly distributed near the river bank, mostly within 500 m from the river channel".

**Reviewer: Page 13: Figure 3 and 4 can be combined**

Authors: Following the reviewer's suggestion, we combined these two figures into Figure 4 (please see Page 17).

Reviewer: Page 14 line 2: We know for what SWC stands for, what are those attached numbers stands for? Ok, it is given in the caption, but, is also needed in the main text.

Authors: As suggested by the reviewer, we added the explanation of SWC when it appears in the main text.

Reviewer: Page 17: the need for Table 3 is clear; why is Table 2 (marginal and conditional effects are not main target of the study)

Authors: Table 2 allows the selection of the key influencing factors from the marginal and conditional effects. Marginal effects reflected the effects of the environmental variable on the community characteristics, while conditional effects reflected the effects of the environmental variables on the community characteristics after the anterior variable was eliminated by the forward selection method. The forward selection in the Table 2 allowed key variables to be determined through the strength of their effects and significance. Based on the key influencing factors selected in the Table 2, we further analyzed the variation of community characteristics explained by different groups of key environmental factors (Table 3). We also revised the title of Table 2. Please see Page 23 Line 22 - Line 23: "The selection of the key influencing factors based on the marginal and conditional effects obtained from the forward selection. Please see Page 23 Line 10 - Line 13: "To further investigate the variation explained by spatial heterogeneity factors and temporal variation factors, we divided those 18 factors into two groups for partitioning analysis (Table 3)".

Reviewer: Page 18 line 17-18: the peculiar result from the vegetation analysis is the bi-modal distribution; do the soil properties show the same pattern; so that to say "variation in soil properties...." Page 19 line 22-24: YES! this can be an inference to the bi-modal distribution (in reference to the above comment)

Authors: We thank the reviewer's suggestion on improving our Discussion on how "variation in soil properties" may affect vegetation community. We developed this topic by referring to the results regarding the variation in soil moisture. Please see Page 25 Line 19 – Line 21: "Although located quite far from the river, soil moisture (e.g., SWC1, SWC2, and SWC3) reached its maximum at 1000 m from river channel (Fig. 6), supporting rich vegetation community (multiple layers of tree-shrub-herb)."

Reviewer: Page 21 line 11-12: Here it says "vicinity to the main roads"? In the methods, it is indicated sampling was done far from roads; explain why?

Authors: In the Data and methods section, the description of "we chose sites that were far from farmlands, roads, irrigated channels and reservoirs" means that we chose plots that were distant from the roads and paths to minimize the human disturbance (i.e., grazing and firewood cutting) on vegetation communities. However, in the study area, there is a main road extending across the oasis and almost parallel to the river channel. As the distance of each sampling plots from the river channel is fixed, it is difficult to avoid sampling near the main road which extents parallel to the river channel. Currently, the vegetation community growing nearby the road is relatively undisturbed as the road is separated from the surrounding by iron wire. Therefore, we believe that sampling near the main road did not go against our general principle about minimizing human disturbance (i.e., grazing and firewood cutting) on the vegetation communities. While the vegetation communities growing nearby the main road (e.g., 1000 m from river channel) might become vulnerable to human disturbance in the future due to increasing population, we described the area that distant 1000 m from river channel as "vicinity to the main roads" in the original manuscript and listed the possible human influence on this gradient in the Discussion section. In the revised manuscript, we explained the information of the main road and clarified our sampling principle in the Data and methods section. Please see Page 9 Line 28 - Page 10 Line 3: "Those sites were far from farmlands, irrigated channels and reservoirs to minimize the impact of human disturbance and other water resources. Although, there is a main road extending across the oasis and almost parallel to the river channel (Fig. 1), the vegetation community growing nearby the road is considered to be undisturbed by the road as the road is separated from the surroundings by iron wire". We added the main road in the Figure 1 to make it easier to be understood (please see Page 8). To avoid further confusion, we rewrote the part in Discussion section and mainly focused on address the potential disturbance near river channel instead of near the road. Please see Page 31 Line 20- Line 23: "In addition to potential threat posed by climate change, the periphery of the river is also more likely to be disturbed by grazing and heavy tourism pressure. Exposure to human disturbance, including trampling by livestock might potentially destroy the soil physical properties, reducing the ecosystem services such as water and soil conservation".

Reviewer: Page 21 line 12-15: To give management options for livestock control; there is a need to have socio-economic background information, specifically to livestock, somewhere in the introduction or in methods.

Authors: Following the reviewer's suggestion, we added the socio-economic background information (e.g. population, farming, tourism) in the Data and methods section. Please see Page 8 Line 1 - Line 6: "The population in the Ejina oasis is 32,410. The local economy mainly depends on the cantaloupe plantation and animal husbandry (e.g., sheep, cattle and

camel). Ejina Oasis is one of China's most important tourist attractions with respect to desert riparian forests, attracting almost 200,000 visitors per year during September to October. Two primary roads are built parallel to the river channel and across the south of the oasis respectively, mainly used for transportation and traveling".

Reviewer: Page 22 line 4: what are "artificial channels"? Or take the whole sentence to Introduction; also line 6, if human disturbance is a problem give a brief background in the Introduction.

Authors: The "artificial channels" are concrete channels built perpendicular to the river with the aim of delivering water for irrigation. They, however, generate little benefit to the surrounding vegetation communities since they lack the seepage property that natural channels have. As we chose our plots that were distant from the roads, farmlands and irrigated channels, human disturbance mentioned in line 6 was not the main factor that shaped the vegetation community, although the extent might increase in the future. To make it easier understand, we replaced the sentence of "artificial channel" with "concrete channel" in the Introduction. Please see Page 3 Line 26 - Line 28: "Every year, about 300 billion m3 of water were delivered using concrete channels that were built perpendicular to the river aiming to expand the river impact and to deliver water for irrigation". We also added the information of possible human disturbance in the Data and methods section. Please see Page 8 Line 1 – Line 6: "The population in the Ejina oasis is 32,410. The local economy mainly depends on the cantaloupe plantation and animal husbandry (e.g. sheep, cattle and camel). Ejina Oasis is one of China's most important tourist attractions with respect to desert riparian forests, attracting almost 200,000 visitors per year during September to October. Two primary roads are built parallel to the river channel and across the south of the oasis respectively, mainly used for transportation and traveling".

Reviewer: Page 22 line 21-26: many more services can be told.

Authors: Following the reviewer's suggestion, we added additional discussion materials regarding ecosystem services, such as sand fixation, carbon storage and water conservation to fully illustrate the importance of conserving the ecosystem. Please see Page 31 Line 16 – Line 23: "Since the influence of ecological water conveyance was mainly limited to 1000 m distance from river, projected rise in temperature could lead to the collapse of riparian vegetation (e.g. *Tamarix ramosissima*, *Lycium ruthenicum*) at further gradients, resulting in decrease of ecosystem service (e.g. sand fixation and carbon storage). In addition to potential threat posed by climate change, the periphery of the river is also more likely to be disturbed by grazing and heavy tourism pressure. Exposure to human disturbance, including trampling

by livestock might potentially destroy the soil physical properties, reducing the ecosystem services such as water and soil conservation."

Jingyi Ding1, Wenwu Zhao1,2\*, Stefani Daryanto2, Lixin Wang2\*, Hao Fan1, Qiang Feng1,3, Yaping Wang1

[revised manuscript text omitted]

---

## Author Response (AR2)

**Responses to the Editor and Reviewers:**

We thank the editor and reviewers very much for the time they spent evaluating our manuscript and providing constructive comments. Their detailed comments helped us to further improve the quality of this manuscript to meet the standard of HESS journal. We have gone through all the comments and amended the original manuscript based on the suggestions and comments. In the following pages we provide point-by-point responses to the editor's and reviewers' comments. Please refer to the attached manuscript with track-changes mode for further details.

**Responses to the Editor:**

Editor: Both of the reviewers found that there had been significant improvements in the manuscript, but that important work remained to be done concerning the quality of the presentation. I am therefore returning the manuscript for further revision so that this new round of comments can be used to bring the standard of the presentation in line with the expectations of the journal.

Authors: We thank the editor for giving us a valuable chance to further improve this manuscript and thank the reviewers for giving details comments and suggestions for this manuscript. We have gone through all the comments and amended the original manuscript based on the suggestions and comments. To improve the quality of presentation, we have carefully condensed and revised the language throughout the manuscript. In addition, the language in our manuscript has been polished by a native English speaker.

Editor: Reviewer #1 has particularly detailed comments that can be of help in addressing issues that detract from reader's ability to readily grasp and appreciate the points being made by the manuscript.

Authors: We thank the Reviewer #1 for giving detailed comments to help us improve the presentation quality of this manuscript. Based on the comments of Reviewer #1, we have carefully revised the manuscript. To improve the readability of this manuscript, the language in our manuscript has been polished by a native English speaker.

Editor: Reviewer #3 has more general comments that will require more work to address, but are still of great value. These include identifying how your work complements or changes the state of the science in relation to what earlier work has shown. And while it is a great step forward to have used remote sensing data to infer the temporal variability in soil moisture, the inferred soil moisture status should be compared to the direct observations of soil moisture and groundwater where available.

Authors: We thank the Reviewer #3 for giving valuable suggestions on this manuscript. We have carefully revised the manuscript according to the comments of Reviewer #3.

To figure out the gaps of studies and highlighted the novelty of our study, we rewrote the

Introduction section. The gaps of previous study were: 1) lack of study that accurately delineate the temporal variation of desert riparian forest; 2) lack of research that incorporates the spatial and temporal variation of desert riparian forest due to the scale inconsistent between spatial and temporal data. The novelty of this study were: 1) accurately delineate the temporal variation of Heihe desert riparian forest along different distances from river channel based on high resolution images; 2) explore both the spatial distribution and temporal variation of Heihe desert riparian forest as well as their influencing factors. Please see Page 5 Line 11-21 in the Introduction section for details: "While large-scale remote sensing data (e.g., MODIS-NDVI, SPOT-NDVI) could capture the general trend of the whole study area, they fail to accurately delineate the temporal variation of desert riparian forest vegetation at the fine scale (i.e.,

Reviewer: 2. The soil moisture and ground water during 2000-2014 were extracted from the retrieved remote sensing data. The results should be validated by comparing with measured data available in some years or some sites. This is important.

Authors: We thank the reviewer for giving us valuable suggestions on the validation of the retrieved remote sensing data. The retrieved remote sensing data was generated by the CLM RIV model, a coupled model that combined the process of stream-aquifer interaction with the Community Land Model Version 4.5 (CLM4.5), based on the high-resolution ASTER DEM dataset, multi-source integrated Chinese land cover map (MICLCover), Heihe watershed allied telemetry experimental research land cover map (HiWATER Land Cover Map), and the China soil characteristics dataset. The providers of the dataset validated this dataset in the published paper (Zeng et al., Hydrology and Earth System Sciences, 2016). Comparisons of simulation outputs and observations from automatic weather station systems and water wells demonstrated that CLM RIV showed considerable ability to reproduce the natural conditions along riverbanks. The soil moisture was well simulated, while the groundwater table was much deeper than the observation groundwater. Based on the validation by the providers, we further corrected the simulated groundwater table by using the monitoring groundwater data along a groundwater monitoring transect (i.e. Wulantuge transect). We added the validation of the retrieved remote sensing data in the Supplement. Please see Page 3-5: "S4 Validation of the retrieved remote sensing data" in the Supplement.

We added the relevant explanation of data validation in the Method section, Please see Page 10 Line 8-9 in the manuscript: "The validation of the retrieved remote sensing data is provided in the Supplement S4". As there were still deviations between simulated data and observed data, we added the possible influence of the accuracy of the remotely sensed soil moisture data on the result of our study in the Discussion section. Please see Page 31 Line 5-8 in the manuscript: "In addition, the deviations between simulations and observations could also partly account for the weak correlations between the variability of soil moisture, groundwater and community characteristics, as the variation of soil moisture and groundwater was derived from the remote sensing dataset."

Reviewer: 3. The Ejina Oasis is mainly subjected to the runoff at Donghe station, not Zhengyixia station which influences the whole downstream of Heihe River Basin.

Authors: We thank the reviewer for the suggestion. We replace the runoff of Zhengyixia

station with the runoff of Donghe station. We made corresponding changes in the Method section. Please see Page 10 Line 20-23 in the manuscript: "The retrieved remote sensing data, monitoring data, land use data, groundwater monitoring data and runoff data at Donghe station (i.e., а hydrological station in the downstream Heihe) (doi:10.3972/heihe.1009.2013.db) were acquired from the Environmental & Ecological Science Data Center for West China, National Natural Science Foundation of China (http://westdc.westgis.ac.cn)". And we recreated the corresponding Figures. Please see Page 22 Figure 6 in the manuscript and Page 8 Figure S6 d in the Supplement.

Reviewer: 4. The variation of groundwater across different distance should be shown in Fig. 6, and incorporated to the analysis of correlation between distribution pattern and environmental factors.

Authors: Following the reviewer's suggestion, we added the spatial variation of groundwater across different distance from river channel. Please see Page 7 Figure S5h in the Supplement. As monitoring the groundwater table in all the sampled sites (35 sites) is difficult, we did not record the groundwater table at all sampling sites in this study. We obtained the groundwater monitoring data at one transect that near the Wulantuge Village (i.e. Wulantuge transect) and used the annual average groundwater during the monitoring period (2010-2012) to depict the spatial distribution of groundwater table along the distance from river channel (Figure S5 h). We added the data resource in the Method section. Please see Page 10 Line 20-23 in the manuscript: "The retrieved remote sensing data, monitoring data, land use data, groundwater monitoring data and runoff data at Donghe station (i.e., a hydrological station in the downstream Heihe) (doi:10.3972/heihe.1009.2013.db) were acquired from the Environmental & Ecological Science Data Center for West China, National Natural Science Foundation of China (http://westdc.westgis.ac.cn)". We added the depiction of groundwater table variation across different distances. Please see Page 6 Line 13-15 in the Supplement: "The annual average groundwater table range from 2.16m near the river bank to 3.27m at the distance of 3000m from river channel. Generally, the groundwater table fluctuated declining with the distance from river."

As we monitoring the groundwater at one transect not all sampling sites, we use the annual average groundwater (GWT\_a) and annual average variability of groundwater (GWT\_c) derived from the retrieved remote sensing data to explore how groundwater influence the vegetation characteristics. Please see Page 21 Table1 and Page 23 Table 2 in Result section and Please see Page 30 Line 30 -Page 31 Line 8 in the Discussion section: "A combination of those aforementioned processes also result in the more stable growth rate of groundwater and soil moisture during 2000-2014 (Fig. S6a, b, c) compared to the runoff fluctuations (Fig. S6d). Therefore, there were weaker relationships between the annual variability in soil moisture and groundwater (e.g., SWC2cm\_c, SWC100cm\_c, GWT\_c) and the annual variability of NDVI (e.g., NDVI\_c) than between the spatial heterogeneity of soil moisture (e.g., SWC3) and the annual variability of NDVI (e.g., NDVI\_c) (Table 1). In addition, the deviations between the variability of soil moisture, groundwater and community characteristics, as the variation of soil moisture and groundwater was derived from the remote sensing dataset."

Groundwater data that derived from retrieved remote sensing data was validated by the sampling data in monitoring data along the Wulantuge transect. Please see in Page 4 Line 17-25 in the Supplement: "In our manuscript, to make the simulated groundwater table fit the actual range of groundwater table in the downstream Heihe, we corrected the simulated groundwater table data by establishing the regression relationship between simulation and observation along Wulantuge transect in the downstream Heihe (Fig. S3). After correction, the corrected groundwater table data (Fig. S4) showed the same temporal variation pattern across different distances from river channel with the simulated data and the groundwater table range from 2.5m-3.5m which was accorded with the range of observed groundwater table (Fig. S5 h). Thus, we use the corrected groundwater table data to analyze the temporal variation of groundwater (Fig. S6 c)."

**Reference:**

[revised manuscript text omitted]
 theof vyegetation composition change inof each community 7 type (I to V) (Fig. 3) was were obtained from the changes in the land use map from 2000 to 2014 (Fig. 8 3). Among the five community types, community V underwent the most changes, with 22.22% of the 9 sites changinged from sparse forest to grassland, 22.22% from grassland to shrubland and 22.22% from 10 bareland to grassland, respectively. The majority (>60%) of the vegetation composition remained 11 unchanged in communitiesy I to IV, with the following exceptions: (i) 37.5% of the sites in community 12 I changed from shrubland to sparse forest and from bareland to grassland, (ii) 33% and 20% of the sites 13 in communitiesy II and III changed from bareland to grassland and from sparse forest to grassland, 14 respectively-, and (iii) 20% of the sites in community IV changed from sparse forest to grassland and 15 another 20% from grassland to shrubland (Fig. 3).

---

## Author Response (AR3)

**Responses to the Editor and the Reviewer:**

We thank the editor and the reviewer for the time they spent evaluating our manuscript and providing constructive comments. Their detailed comments helped us to further improve the quality of this manuscript. We have gone through all the comments and modified the original manuscript based on the suggestions and comments. In the following pages we provide point-by-point responses to the editor's and the reviewer's comments. Please refer to the attached manuscript with track-changes mode for further details.

**Responses to the Editor:**

Editor: Referee number 4 has provided a set of comments that can further improve the manuscript. While I have essentially recommended publication, I urge the authors to make use of referee #4 ś comments as a minor revision.

Authors: We thank the editor for giving us a valuable chance to further improve this manuscript and thank the reviewer#4 for giving detailed comments and suggestions for this manuscript. We have gone through all the comments and amended the original manuscript based on the suggestions and comments.

**Responses to the Reviwer#4:**

**Major issues:**

Reviewer: While this manuscript (MS) is interesting and at this stage pretty well written, there are still several ambiguities and shortcomings. After I read the MS I went back to the first round of reviews to see what comments other referees had, and to my surprise I recognized some of my own comments there. That is, some comments from the first round are still valid and have not been properly addressed.

Authors: We thank the reviewer for giving us constructive suggestions and detailed comments to improve the quality of this manuscript. Following the reviewer's comment on the unaddressed comments during the first round of review in this manuscript, we have carefully checked the comments and amended these unaddressed comments in the manuscript (e.g., the definition of variables and the description of data collection; please see refer to the responses to comments below). We have further revised the language throughout the manuscript and the supplementary material to avoid the ambiguities in expression.

Reviewer: All in all, the language in the main text of the MS is fine now (though I found some wordings that sound strange to me, especially in the Introduction, and the language of the Supplementary, especially S4, needs improvement), but the presentation is still unclear in certain parts of the MS. For instance, in the Discussion it was at times not clear to me what are results/conclusions of this work and what is conclusions from the literature (this comment also applies to parts of the Introduction). This may be a language issue, since past tense is used throughout, even for previously published results. This usage gives the impression that the authors are talking about their results although they are actually discussing results from the literature. An example: "Bulk density accounted for 13.51% of the total explanation of the vegetation variance and was negatively correlated with the annual average NDVI (NDVI_a) and the annual variability of NDVI (NDVI_c). In the hyperarid zone, soils with high bulk density were often characterized as having a high proportion of coarse soil particles but low clay content" (P21, lines 18-22). I guess the first sentence here refers to results from this study whereas the second sentence refers to the literature. However, the usage of "were" instead of "are" in the second sentence give the impression that this is a result of the present study (though this has not been shown in the MS). This kind of wording is very confusing, but common throughout the Discussion (see also e.g. P19, lines 12-19) and Introduction.

Authors: We thank the reviewer for pointing out the language issues in the manuscript and the supplementary material. We have carefully revised language throughout the manuscript and the supplementary material, especially the Introduction and Discussion section in the manuscript and S4 in the supplementary material.

Based on the reviewer's comments on the unclear presentation in Introduction and Discussion section, we use the present tense when depicting the results from literature and we added the depiction such as "Study shows that" when refer to the literature throughout the manuscript and the supplementary material. As to the example "P21 lines 18-22" put by the reviewer, please see the revised sentences in the manuscript at Page 24 Line 16-21: "Bulk density accounted for 13.51% of the total explanation of the vegetation variance and was negatively correlated with the annual average of NDVI (NDVI_a) and the annual variability of NDVI (NDVI_c). Study shows that, in the hyperarid zone, soils with high bulk density are often characterized by a high proportion of coarse soil particles but low clay content (Ravi et al., 2009)." As to the example "P19, lines 12-19" put by the reviewer, please see the revised sentences in the manuscript at Page 21 Line 28- Page 22 Line 10: "At the same time, other study find that the presence of the deep-rooted tree, *Populus euphratica*, can redistribute deep soil water to the shallow layer as a strategy of mutualism, benefiting the growth of shallow-rooted herbaceous species (Hao et al., 2013). Similar mechanism also occurs at further distance (i.e., 3000 m) from the river. Although situated in the transition region (from riparian forest to desert shrubs), soils at the distance of 3000 m from river channel were still rich in fine particles (clay and silt; 35.6%) (Table S3), brought by the interaction between wind erosion and shrubs (Ravi et al., 2009). The presence of fine soil particles can increase the soil water holding capacity and soil nutrients around the shrub patches ('fertile islands') (Ravi et al., 2010), allowing the growth of some xerophytic herbs and increasing the community diversity in arid region (Stavi et al., 2008)".

Reviewer: Some of the sampling procedures and monitoring require better descriptions (and this was one of the comments from the first round of reviews). For instance, how were the vegetation quadrats chosen? Randomly? How does the groundwater monitoring transect look like (e.g. how many wells, how deep were the wells, what was the precision and accuracy of the HOBOs and at what distances from the river were the wells located)? Same thing with soil moisture monitoring – where was this done? At how many sites? And where were the sites located? Adding the locations of these monitoring sites to the map would be helpful.

Authors: Following the reviewer's suggestion, we added all the descriptions of the sampling procedures and monitoring data in the Methods section. The vegetation quadrats were chosen randomly. We added the description in the Method section. Please see Page 8 Line 8-9 in the manuscript: "Three tree quadrats (30 m × 30 m) and shrub quadrats (10 m × 10 m) were established randomly at each site." In addition, we also added the detailed information on the acquisition of soil properties. Please see Page 8 Line 20-30 in the manuscript: "Soil samples were collected every 20 cm at a depth of 100 cm in each site to determine the soil composition and chemical properties. Surface soil samples (from a depth of 0–20 cm) were subsequently analyzed in the laboratory to determine their clay (<0.002 mm), silt (0.002–0.05 mm), sand (0.05–2 mm), and gravel (>2 mm) content using a Malvern Mastersizer 2000. Soil organic matter (SOM) was measured using the $K_2Cr_2O_7$ method (Liu, 1996). Total nitrogen (TN) was determined using the Kjeldahl procedure (ISSCAS, 1978). Total phosphorus (TP) was determined using a UT–1810PC spectrophotometer (PERSEE, Beijing, China), after $H_2SO_4$–$HClO_4$ digestion (ISSCAS, 1978). Total salt content (TS) was determined by oven method (Liu, 1996)".

The detailed information of the groundwater monitoring points (including the locations of wells and accuracy of HOBOs) was added in the Methods section. Please see Page 9 Line 17-23 in the manuscript: "The spatial variation of groundwater table was obtained from groundwater monitoring data recorded by seven wells (i.e., 7.62–9.66 m deep) in the Ejina oasis, located at 50 m, 300 m, 2200 m, 2700 m, 3200 m, and 3700 m from the river center (Fig. 1). These monitoring data of the groundwater table were recorded as 18-hour averages with a three decimal places accuracy using a HOBO automatic groundwater table gauge from October 2010 to December 2014 (Fu et al., 2014)".

We also added the following description of soil moisture monitoring point. Please see Page 9 Line 23-28 in the manuscript: "The diurnal and annual variations of soil moisture were obtained from the monitoring data of soil moisture, recorded at 0.5 Hz, as a 10-min average from 2013–2015 using a suite of micrometeorological sensors (CR800, CR23X, CR23XTD, Campbell Scientific Inc.) installed at a site that is located within Heihe riparian forest, about 1500 m from the Heihe river channel (Fig. 1) (doi:10.3972/hiwater.241.2015.db; doi:10.3972/hiwater.318.2016.db) (Liu et al., 2011)".

In addition, we added locations of these monitoring points in the map of study area (Fig. 1B). Please see Page 7 Figure 1 in the manuscript.

Reviewer: I am not familiar with all the indices used in the manuscript but I found the description of some of them incomplete. For example, how are the variables/parameters in equations 1 and 2 defined? Also, could you please cite a source for this method? What does A and Ai stand for in equation 4? What do you mean by "importance value" (P. 10, line 26)? The variables TN, TP and TS are never defined – what do they stand for (again, this was a comment by one of the first referees)? How is riparian zone defined? Based on distance, geomorphology, soil properties, topography or something else?

Authors: We thank the reviewer for pointing out the incomplete presentation. We have carefully revised these problems and checked the similar issues throughout the whole manuscript.

Following the reviewers' suggestion, we added the explanation and reference resource on the variables in equation1 and 2. Please see Page 10 Line 10-11 in the manuscript: "where P $_{Tree}$ is the importance value in the tree layer. P $_{Shrub\ or\ Grass}$ is the importance value in the shrub or grass layer". A reference for the method (i.e. Zhang, 2011) was also added in the manuscript (Page 10 Line 7).

$A$ is the total diversity index of the community; $A_i$ is the diversity index of the $i^{th}$ growth type. We added the explanation in the manuscript. Please see Page 10 Line 27- Page 11 Line 1 in the manuscript: "where $A$ is the total diversity index of the community; $i = 1$ for the tree layer, 2 for the shrub layer and 3 for the herb layer; $W_i$ is the weighted parameters of the diversity index for the $i^{th}$ growth type; $A_i$ is the diversity index of the $i^{th}$ growth type". In addition, we also revised the definition of variables in equation 5-8. Please see Page 11 Line 12-13 in the manuscript: "where $P_i$ is the relative importance value of the $i^{th}$ species, and $S$ is the total number of species in each growth type at each sampling site".

The importance value is an index that depicts the relative importance of plant species in the community. We added the explanation in the manuscript. Please see Page 10 Line 4-7 in the manuscript: "The importance value (P), an index that characterizes the relative importance of plant species in the community (Zhang and Dong, 2010) was calculated for each species at each tree shrub and herb layer in every sampling site using the following formulas (Zhang, 2011)".

We apologize for not defining the variables TN, TP and TS in the main text of the manuscript. We added the definition of these variables when it first appeared in the manuscript. Please see Page 8 Line 26-30 in the manuscript: "Soil organic matter (SOM) was measured using the $K_2Cr_2O_7$ method (Liu, 1996). Total nitrogen (TN) was determined using the Kjeldahl procedure (ISSCAS, 1978). Total phosphorus (TP) was determined using a UT–1810PC spectrophotometer (PERSEE, Beijing, China), after $H_2SO_4$–$HClO_4$ digestion (ISSCAS, 1978). Total salt content (TS) was determined by oven method (Liu, 1996)." We carefully checked the similar issues throughout the manuscript. We revised the variables in the note of table. Please see Page 21 Line 1-7 in the manuscript: "a: spatial distribution factors, including 0–30 cm soil moisture (SWC1), 30–100 cm soil moisture (SWC2), 100–200 cm soil moisture (SWC3), bulk density (BD), clay, silt, sand, gravel, soil organic matter (SOM), total nitrogen (TN), total phosphorus (TP), total salt content (TS); b: temporal factors, including annual average of groundwater table (GWT_a), annual average of 2 cm soil moisture (SWC2cm_a), annual average of 100 cm soil moisture (SWC100cm_a), annual variability of groundwater table (GWT_c), annual variability of 2 cm soil moisture (SWC2cm_c), annual variability of 100 cm soil moisture (SWC100cm_c)". We also added the definition of the diversity indices when they first appeared in the manuscript. Please see Page 11 Line 3-10 in the manuscript: "Species diversity indices were calculated (Liu et al., 1997), including the Shannon-Wiener diversity index ($H$); Simpson dominance index ($D$) was calculated as; and Pielou evenness index ($J_{sw}$) was calculated as; Finally, Patrick richness index ($R$) was calculated as". In addition, we revised the definition of variables in the Supplementary

Material S5. Please see Page 6 Line 13-16 in the Supplementary Material: "SOM (soil organic matter), TN (total nitrogen) and TP (total phosphorus) generally decreased along the distance from river channel and reached a relatively high value at the distances of 500 m and 2000–2500 m. TS (total salt content) peaked at the distance of 1000 m (2.57%) (Table S3) and dropped gradually".

We added the definition of the riparian zone in the Introduction section. Please see Page 2 Line 23-27 in the manuscript: "Riparian zone is the linkage between terrestrial and aquatic ecosystems (Naiman and Décamps, 1997), which is usually defined as the stream channel between the low- and the high-water marks, in addition to the terrestrial landscape above the high-water mark. Consequently, vegetation in the riparian zone is likely to be influenced by elevated water tables or extreme flooding and by the ability of soils to hold water (Nilsson and Berggren 2000)." We also completed the explanation on how we choose the sampling range of the riparian zone in the downstream Heihe. Please see Page 7 Line 8-14 in the manuscript: "However, the spatial extent of the riparian zone is difficult to precisely delineate due to the heterogeneity of the landforms (Décamps et al., 2004). Although previous studies indicates that the forests are distributed between 0 m and 2000 m from the river channel, corresponding to the influence range of ecological water conveyance (Si et al., 2005), our study extended beyond that range (i.e., up to 3000 m from the river channel) to fully cover the distribution pattern of the desert riparian forests in downstream Heihe River".

Reviewer: Not all data are reported. The soil characteristics, which are key to the paper, are never reported. Why not add a table with those data?

Authors: We thank the reviewer for this suggestion. We added the Table S3 to depict the characteristics of environmental factors (including soil characteristics and groundwater) at different distances from the river channel. Please see Page 8 Table S3 in the Supplementary Material: "The characteristics of environmental factors at different distances from river channel".

Reviewer: Some of the supplementary material is not referred to in the main text. If it is essential for the MS, it should be referred to.

Authors: We thank the reviewer for the suggestion. We have carefully checked and added the reference of each supplementary material. We added the reference of Figure S2 in the manuscript. Please see Page 11 Line 17-19: "The monitoring data of soil moisture in desert riparian forest showed that soil moisture under 20 cm was relatively stable and could represent water condition at the sampling site (Fig. S1)." We added the reference of Table S2 in the manuscript. Please see Page 11 Line 14-15: "The least significant difference (LSD) test was used to determine the significance of the variability in vegetation characteristics among five transects (Supplementary Material S3, Table S2)." We added the reference of Table S3 in the manuscript. Please see Page 22 Line 4-7: "Although situated in the transition region (from riparian forest to desert shrubs), soils at the distance of 3000 m from river channel were still rich in fine particles (clay and silt; 35.6%) (Table S3), brought by the interaction between wind erosion and shrubs (Ravi et al., 2009)." We added the reference of Figure S7 in the manuscript. Please see Page 20 Line 5-7: "When combined, these factors explained 15.2% of the community characteristics variation, accounting for 34.3% of the total explanation that was explained by all the investigated environmental factors (Fig. S7)".

Reviewer: Based on the history of this MS I perhaps should recommend rejection, especially since not all previous comments have been addressed. However, the main story is interesting (though I have to admit that the topic is far beyond my own competence, not being a desert ecologist). I therefore suggest major revisions of the MS before the editor should consider it for publication.

Authors: We thank the reviewer for giving us a valuable chance to further improve this manuscript. We have amended the original manuscript based on all the suggestions and comments. Following the reviewer's comment on the unclear presentation, we have carefully checked and revised the similar issues throughout the manuscript and the supplementary material. In addition, we have carefully checked the comments and amended these unaddressed comments in the manuscript and the supplementary material (e.g., the definition of variables and the description of data collection; please refer to the responses to the major issues above).

**Minor issues:**

Reviewer: Detailed comments (these are only the most important minor comments I found)

Authors: We thank the reviewer for giving us these valuable and detailed comments, we have carefully revised them and checked the similar issues throughout the whole manuscript.

Reviewer: P3, line 2: what do you mean by "diversity level"? Biodiversity?

Authors: We replaced the "diversity level" with "biodiversity". Please see Page 3 Line 5-7 in the manuscript: "However, due to their low biodiversity and weak resilience, desert riparian forests are sensitive to disturbance and likely to be threatened by climate change and human disturbance ( Li et al., 2013)".

Reviewer: P3, line 13: What oasis area? Do you refer to a specific oasis or oases in general?

Authors: It referred to the oasis area in the downstream Heihe. To make it clear, we revised this sentence. Please see Page 3 Line 17-21 in the manuscript: "While most of vegetation in the downstream Heihe River has been restored ( Lü et al., 2015), nearly 20% of its oasis area covered by desert riparian forests remains degraded despite better downstream water conditions and the rising of groundwater table (Zhang et al., 2011a)".

Reviewer: P3, lines 18-29: I found this section confusing. You start off with general statements but then suddenly you refer to specific results. It was not clear to me what refers to desert forests in general and what is specific info about desert forests in the Heihe Basin.

Authors: We thank the reviewer for pointing this out. To make it clear, we carefully rewrote this section. Please see Page 3 Line 29- Page 4 Line 15 in the manuscript: "In hyperarid zone, groundwater is regarded the major driving factor of vegetation distributions and groundwater table should be between 2 m and 4 m deep to support vegetation growth (Zheng et al., 2005). Study in Tarim River, for example, shows that species diversity peaks where groundwater depth is around 2–4 m. Once groundwater tables falls beyond 4.5 m, a deficiency in soil moisture occurs, followed by degradation of vegetation communities (Li et al., 2013). While groundwater drops rapidly away from the river bank to approximately 6 m deep at a distance of 1000 m from Tarim River channel, (Aishan et al., 2013), groundwater table in downstream Heihe River, where most of the desert riparian forest are distributed, remains above 4 m even at the distance of 3800 m from the river channel (Wang et al., 2011; Fu et al., 2014). Yet some sites have not been completely restored in the Heihe riparian zones, and its downstream vegetation community is still dominated by shrubs instead of multiple layers of tree (He and Zhao, 2006; Zhang et al., 2011a)."

Reviewer: P4, line 5-6: Does soil moisture affect hydrological processes? Or are hydrological processes affecting soil moisture?

Authors: We apologize for the confusion of this sentence. To make it clear, we revised this sentence. Please see Page 4 Line 20-22 in the manuscript: "Apart from groundwater, soil properties, such as soil moisture, soil physical and chemical properties also shape the community characteristics and species vitality (Stirzaker et al. 1996; Salter and Williams, 1965)".

Reviewer: P4, line 9: Are you talking about diversity or abundance? Most of the MS is about diversity so I find it strange that you talk about abundance here.

Authors: Species abundance is actually only one of the indices to depict the diversity of the community. To make the sentence easier to understand, we replaced the "abundance" with "diversity" and revised the whole sentence. Please see Page 4 Line 24-28 in the manuscript "As different depths of soil moisture affect species diversity in each community layer (D'Odorico et al., 2007; Fang et al., 2016), a decline in soil moisture may reduce the diversity of drought-sensitive species (e.g., herbs), resulting in a community shift towards drought-tolerant vegetation types with distance from the river channel (Zhu et al., 2014)".

Reviewer: P4, line 12: Changes due to what?

Authors: In this sentence we cited the result of a research conducted in the hyperarid zone in the Tarim River to illustrate the role of soil nutrient in influencing the community dynamics. Based on grey correlation analysis, the research found that there exist significant relationship of plant species diversity with soil nutrient which suggesting that changing in soil nutrient could accounted for the changes in species diversity in the lower reaches of Tarim River. This research mainly focused on investigating the driving forces for the changes in species diversity and the reasons for changes in soil nutrient were not addressed in this research.

Reviewer: P5, line 6-8: I do not think you ever address the second part of this objective, and I do not think that you should either. I suggest you remove it; you do not present any results related to this objective ("appropriate restoration and protection measures … under a changing environment").

Authors: Following the reviewers' suggestion, we deleted this part of objectives. Please see Page 5 Line 23-25 in the manuscript.

Reviewer: P5, line 8: I suggest you use "studied" or "sampled" instead of "used".

Authors: Following the reviewer's suggestion, we replaced the word "sampled" with "studied". Please see Page 5 Line 25-28 in the manuscript: "We studied 3000 m transects running perpendicular to the river channel to include different distances from the river channel and to depict the spatial distribution of vegetation (e.g., changes in floristic composition, community structure and diversity) at each distance from the river channel".

Reviewer: P5, line 11: What sampling? You have not described any sampling thus far.

Authors: According to the reviewer's comments, we revised this sentence and deleted the "field sampling". Please see Page 5 Line 28-30 in the manuscript: "We used NDVI variations at different distances from the river channel, derived from high resolution images (e.g., 30 m resolution) from 2000–2014, to depict the temporal variation of the vegetation".

Reviewer: P6, line 5: Roads are used for transport and travel – remove this sentence, it is a tautology

Authors: Following the reviewer's suggestion, we deleted this sentence. Please see Page 6 Line 20 in the manuscript.

Reviewer: P6, line 11: I don't think these are formal soil types according to common soil classification schemes (e.g. FAO)

Authors: According to the reviewer's comments, we replaced these soil types with formal soil types. As the soil types in the downstream Heihe are intensely regional, we used formal soil types according to the Chinese soil taxonomic classification scheme referred from the book of Chinese soil geography (Gong, 2014). Please see Page 6 Line 25- Page 7 Line 1 in the manuscript: "The main soil types in the area are Gypsi Sali–Orthic Aridosols and Calcaric Aridi–Orthic Primosols. Para–alkalic Aqui–Orthic Halosols and Calcaric Ochri–Aquic Cambosols also exist in the lake basins and lowlands (Chen et al., 2014; Gong., 2014)".

Reviewer: P6, Figure 1. The second sentence is repetition from the main text – is it necessary?

Authors: Following the reviewer's suggestion, we deleted this sentence. Please see Page 7 Line 4 in the manuscript.

Reviewer: P8, line 8: 1000 m sound pretty coarse to me – what is the cause of this resolution?

Authors: In this study, temporal variation of soil moisture and groundwater were used to investigate the impact of water variability on the desert riparian forest. As there was lack of the long-term field monitoring data during 2000-2014, we used the remotely sensed dataset with 1000 m resolution to derive the temporal variation of 2 cm soil moisture, 100 cm soil moisture and groundwater at each site during the research period. The retrieved remote sensing data was generated by the land model CLM4.5 using the high-resolution ASTER DEM dataset and the multi-source integrated Chinese land cover map (MICLCover). As the resolution of Chinese land cover map (MICLCover) was 1000m, the whole retrieved remote sensing dataset was in 1000m resolution. Although the resolution of this dataset was quite coarse compared to the field sampling data, it was characterized with high temporal resolution and its temporal variation pattern could well depict the water variability during the research period. Thus we used this dataset to depict the temporal variation of water variability and its influence on the desert riparian forest. As to the accuracy of this remote sensing dataset, please see "S4 Validation of the retrieved remote sensing data" in the supplementary material.

Reviewer: P 11, line 1: what is a "contribution rate"?

Authors: To make it easier to understand, we rewrote this sentence and replaced the "contribution rate of each factor" with "the percentage of vegetation variance explained by each factor". Please see Page 12 Line 13-16 in the manuscript: "To further separate the key influencing factors of the 18 environmental variables, the marginal and conditional effects of the various variables were calculated through Monte Carlo forward selection in RDA (redundancy analysis), which directly showed the significance and the percentage of vegetation variance explained by each factor".

Reviewer: P15, lines 18-19: this is really methods description

Authors: According to the reviewer's comments, we deleted these methods description and revised the sentence. Please see Page 17 Line 16-19 in the manuscript: "The correlation coefficient between NDVI and runoff was measured to examine the relationship between runoff and NDVI (Fig. 6)".

Reviewer: P17, lines 11-13: this is really methods description

Authors: According to the reviewer's comments, we deleted these methods description and revised the sentence. Please see Page 19 Line 12-13 in the manuscript.

Reviewer: P17, lines 13-17: I do not understand all these percentages – could you clarify? Also, I guess you should refer to Fig S7 here.

Authors: These percentages represented the percentage of community characteristics variation explained by different groups of factors and the percentage they account for in the total explanation that explained by all the investigated environmental factors. To make this section easier understand, we rewrote these sentences and clarify the meaning of these percentages. Please see Page 19 Line 13- Page 20 Line 7 in the manuscript: "Spatial heterogeneity factors explained 43.5% of the variation of community characteristics and accounted for 98.4% of the variance explanation explained by all the investigated environmental factors (Table 3). This result indicated that spatial heterogeneity of environmental factors was the major driving force of the spatio-temporal variation in this desert riparian forest. In contrast, temporal variation factors only explained 15.9% of the variation of community characteristics, accounting for 35.9% of the total explanation that was explained by all the investigated environmental factors (Table 3). This result suggested that temporal variation of the environmental factors exerted less impact on community characteristics compared to spatial heterogeneity of environmental factors in this desert riparian forest. While each factor group affected community characteristics separately, the spatial heterogeneity factors and temporal variation factors also jointly shaped the variation of community characteristics in downstream Heihe riparian forest. When combined, these factors explained 15.2% of the community characteristics variation, accounting for 34.3% of the total explanation that was explained by all the investigated environmental factors (Fig. S7)".

Reviewer: P19, line 10: what does "this region" and "They" refer to?

Authors: To make it clear, we revised these sentences. Please see Page 21 Line 24-28 in the manuscript: "We attributed fine-textured soils found in the upper soil layer to the high species diversity at sites located about 1000 m from the river channel. Previous study shows that fine-textured soils increase the soil water holding capacity and improve the gravimetric moisture content in the upper soil layer, which provide suitable habitat for the growth of diverse herb species with shallow rooting systems (Liu et al., 2008)".

Reviewer: P19, line 12-19: Are these conclusions based on this study? I do not think it is possible to draw these conclusions based on the data from this study. The wording is confusing.

Authors: These conclusions were drawn based on literature study. To make it clear, we used present tense and added "study finds that" when referring to the results from literatures. Please see the revised sentences in the manuscript at Page 21 Line 28-Page22 Line10: "At the same time, other study finds that the presence of the deep-rooted tree, *Populus euphratica*, can redistribute deep soil water to the shallow layer as a strategy of mutualism, benefiting the growth of shallow-rooted herbaceous species (Hao et al., 2013). Similar mechanism also occurs at further distance (i.e., 3000 m) from the river. Although situated in the transition region (from riparian forest to desert shrubs), soils at the distance of 3000 m from river channel were still rich in fine particles (clay and silt; 35.6%) (Table S3), brought by the interaction between wind erosion and shrubs (Ravi et al., 2009). The presence of fine soil particles can increase the soil water holding capacity and soil nutrients around the shrub patches ('fertile islands') (Ravi et al., 2010), allowing the growth of some xerophytic herbs and increasing the community diversity in arid region (Stavi et al., 2008)" .

Reviewer: P20, line 4: No results presented to show this

Authors: This conclusion was drawn based on a literature study. To make it clear, we revised this sentence. Please see Page 22 Line 22-26 in the manuscript: "As the fine roots of most herb species are mainly distributed within the top 30 cm of the surface soils (Fu et al., 2014), surface soil moisture (0–30 cm soil moisture; SWC1) likely became the main water source for the herb layers and contributed to a high coverage of dominant herb species, such as *S. alopecuroides* and *K. caspica*".

Reviewer: P20, line 14: No results presented to show this

Authors: This conclusion was drawn based on a literature study. To make it clear, we revised this sentence. Please see Page 23 Line 6-9 in the manuscript: "Bulk density and soil composition (clay, silt, gravel), which are critical for water and nutrient holding capacity (Stirzaker et al., 1996; Meskinivishkaee et al., 2014), mainly influenced community density, community coverage, shrub coverage and diversity indices (Table 1)".

Reviewer: P23, lines 15-19: I do not think this paragraph is necessary and suggest that you remove it.

Authors: We thank for the reviewer's suggestion. However, we consider this paragraph necessary as it provides the reason for conducting this study. Since water availability and soil physical properties were the key factors influencing community structure in this desert riparian forest, we proposed to better protect soil conditions under climate change and intensive human activities. To illustrate the necessity of this paragraph, we revised these sentences. Please see Page 26 Line 22-29 in the manuscript: "Our study showed that water availability and spatial heterogeneity of soil properties were the main driving forces for the spatial distribution and temporal variation of restored desert riparian forest at Heihe River Basin. To halt degradation in this critical zone, we suggested to build natural channels perpendicular to the river to fully extend the influencing scope of the ecological water conveyance and benefit the regions far from the river bank (Zhang et al., 2011b). At the same time, multiple conservation measures such as establishing critical fenced areas for ecological protection and constructing artificial shields or establishing straw checker boards on the bare land to prevent erosion, were recommended around the periphery of the river".

Reviewer: P23, lines 26-27: This has not been mentioned before. I guess you could describe the ecological water conveyance better in the Introduction

Authors: Following the reviewer's suggestion, we added the influencing range of ecological water conveyance to the Introduction section. Please see Page 3 Line 16-17 in the manuscript: "The influence of ecological water conveyance may reach as far as 2000 m from the river (Si et al., 2005; Zeng et al., 2016)".

Reviewer: P24, lines 1-5: I do not think this paragraph is necessary and suggest that you remove it.

Authors: We thank the reviewer's suggestion. However, we consider this sentence necessary as it provides some suggestions to improve the current restoration efforts in this desert riparian forest. To make it clear, we rewrote these sentences. To make it clear, we shorted these sentences. Please see Page 27 Line 10-13 in the manuscript: "
[revised manuscript text omitted]